# Uni-MDTrack: Prompt Unified Single Object Tracking with Deep Fusion of Memory and Dynamic State

## Abstract

In this paper, we propose a simple but powerful parameter-efficient fine-tuning (PEFT) framework designed for unified single object trackers. Our framework is built upon two novel components: a Memory-Aware Compression Prompt (MCP) module and Dynamic State Fusion (DSF) modules. MCP effectively compresses memory features into memory-aware prompt tokens, which are deeply interacted with the input sequence throughout the entire backbone, significantly enhancing model performance while maintaining a stable computational load. DSF complements the discrete memory features by capturing the continuous dynamic state of the target, progressively introducing the updated dynamic state features from shallow to deep layers of the tracker, while also preserving high operational efficiency. MCP effectively overcomes the limitations of previous trackers that rely on only a few frames when introducing memory, which significantly increases input length and computational cost. It also addresses the insufficient fusion problem in existing memory-prompting methods. DSF remedies the lack of dynamic feature about continuous target variation in prior PEFT methods. Based on the MCP and DSF modules, we propose Uni-MDTrack, a tracker that supports tracking across five modalities. Experimental results across 10 datasets spanning five modalities demonstrate that Uni-MDTrack achieves state-of-the-art performance, with only 30% of parameters requiring training. Furthermore, both MCP and DSF exhibit excellent generality, functioning as plug-and-play components that can boost the performance of various trackers. Code will be released for further research.

## 1 Introduction

Modern SOT methods (Ye et al., 2022; Zhu et al., 2023; Cai et al., 2024; Hong et al., 2024; Lin et al., 2025) adopt one-stream paradigm, leveraging Transformer-based backbones (Dosovitskiy et al., 2021) to process both the template and search region simultaneously using self-attention. Despite the strong capacity for template-search region feature extraction and relation modeling, one-stream trackers incur substantial training costs, demanding large-scale datasets and extensive training epochs. Consequently, a growing number of methods adopt RGB-based trackers trained from scratch as the pre-trained foundation model, and focus on boosting performance through parameter-efficient fine-tuning (PEFT).

The first category of PEFT methods augments RGB foundation trackers with auxiliary modalities such as infrared, depth, event, or language, by extracting and integrating features into foundation trackers via lightweight trainable modules (Zhu et al., 2023; Hou et al., 2024; Hu et al., 2025; Wu et al., 2024). However, after fine-tuning, such models can no longer maintain high performance RGB-only tracking. Meanwhile, with the expansion of multimodal visual tracking data scales, unified trackers, such as SUTrack (Chen et al., 2025) and FlexTrack (Tan et al., 2025), which are directly trained on various modality datasets, can support tracking in all modalities and achieves state-of-the-art performance in multimodal tracking. Therefore, given the emergence of foundation trackers with unified multimodal tracking capabilities, our work focuses more on the second category of PEFT methods: enhancing the foundation model by introducing additional lightweight trainable modules, particularly enhancing the spatio-temporal context modeling capability (Lin et al., 2025; Cai et al., 2024; 2025), as shown in Figure 1.

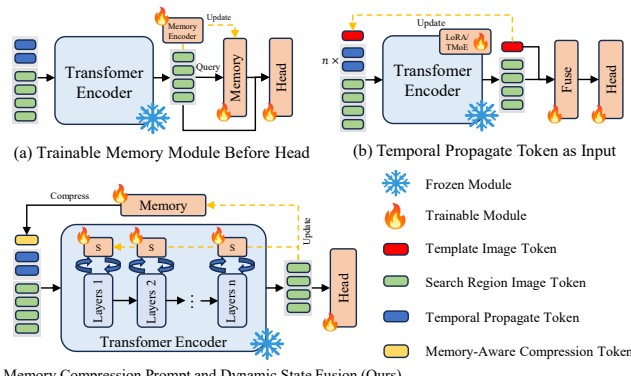

(a) Trainable Memory Module Before Head

(b) Temporal Propagate Token as Input

(c) Memory Compression Prompt and Dynamic State Fusion (Ours)

Figure 1: Comparison of PEFT strategies for enhancing spatio-temporal modeling in a foundation tracker. (a) Introducing memory features and fusing them before the prediction head. (b) Temporal propagation tokens and auxiliary templates. (c) Our approach: Memory compression prompts for the entire model, combined with multi-level dynamic state integration.

Prior typical work (Cai et al., 2024) demonstrates that providing a foundation tracker with spatio-temporal features leads to significant performance gains, and introduces a memory-based prompt module, which achieves substantial performance improvements while keeping the backbone network frozen, as shown in Figure 1(a). However, memory-based methods update the memory at fixed frame intervals, and cannot effectively handle drastic, short-term target variations. And the memory features are not introduced until the prediction head, which results in a lack of deep fusion with the search region features.

SPMTrack (Cai et al., 2025) achieves the current state-of-the-art performance on RGB-based tracking by fine-tuning with TMoE and temporal propagation token, as shown in Figure 1(b), while uniformly sampling several auxiliary templates from historical frames to incorporate long-term memory. Many from-scratch training methods also adopt auxiliary templates and temporally propagated token. However, these components have inherent limitations. The propagated token interacts with both the template and search region tokens simultaneously, which leads to a significant portion of its attention being focused on the template area, making it function more as a template enhancer rather than a representation of continuous state changes of the target. Furthermore, the sparsely sampled auxiliary templates neglect a vast amount of contextual information and risk introducing distractors, while substantially increasing computational costs due to the extended input length. These limitations motivate our question: *How to deeply integrate rich memory and robust dynamic features of the target into a foundation tracker in a parameter-efficient manner?*

To address the above issues, in this paper, we propose a prompt module based on memory compression tokens (MCP) and a dynamic state fusion module based on State-Space Models (DSF). As shown in Figure 1(c), similar to memory-based methods, MCP also maintains a memory bank. The key distinction lies in that MCP employs dynamic queries to compress memory bank into a fixed set of memory-aware tokens. By concatenating these tokens to the input sequence, MCP achieves deep interaction between memory, template and search region features with a minimal increase in sequence length, thereby preserving computational efficiency. Meanwhile, memory-aware tokens also alleviate the problem of limited contextual information contained in temporal propagated tokens. DSF performs continuous updates of the target state based on a State Space Model (SSM). DSF only uses search region features to update the state to ensure sufficient capture of the dynamic changes of the target. Moreover, DSF employs a multi-level shallow-to-deep fusion strategy, integrating dynamic state feature throughout the backbone.

Based on our proposed MCP and DSF modules, we present Uni-MDTrack, a novel tracker that demonstrates remarkable training efficiency by fine-tuning only the MCP, DSF, and prediction head. We use data across 5 modalities for training: RGB, RGB-T, RGB-D, RGB-E, and RGB-Language. Training under 30% of its parameters for only 50 epochs, Uni-MDTrack achieves state-of-the-art performance on 10 datasets including LaSOT (Fan et al., 2019), TrackingNet (Muller et al., 2018), VisEvent (Wang et al., 2024), and Depthtrack (Yan et al., 2021b). Moreover, our proposed MCP and DSF modules demonstrate excellent generalization ability, acting as plug-and-play enhancements that effectively elevate the performance of diverse trackers.

To summarize, our contributions are as follows: **(1)** We propose a prompt module based on memory compression tokens (MCP) and a dynamic state fusion module based on SSMs (DSF) to efficiently and robustly introduce target spatio-temporal context features and continuous dynamics. **(2)** We propose Uni-MDTrack, a novel tracker that efficiently and deeply integrates the features from MCP and DSF modules, while retaining only 30% of the trainable parameters. **(3)** Experimental

results demonstrate that Uni-MDTrack achieves state-of-the-art performance on 10 datasets across five modalities. Furthermore, our proposed MCP and DSF modules can be used as plug-and-play components to effectively enhance the performance of various other models.

## 2 RELATED WORK

### 2.1 PARAMETER-EFFICIENT FINE-TUNED TRACKERS

Most current trackers adopt one-stream paradigm (Ye et al., 2022; Cui et al., 2022; Wu et al., 2023; Chen et al., 2022). However, one-stream trackers demand significantly more training steps and employ large backbones such as ViT-L (Dosovitskiy et al., 2021), leading to a substantial training costs. Thanks to abundant RGB data and the class-agnostic nature of the SOT task, trackers possess strong generalization capabilities. Consequently, a growing body of research has shifted towards PEFT to enhance the performance of existing trackers. PEFT methods can be broadly divided into two categories. The first category introduces auxiliary modalities to supplement an RGB-based foundation tracker (*e.g.* RGB-D, RGB-T, RGB-E). ProTrack (Yang et al., 2022), ViPT (Zhu et al., 2023), and SeqTrackV2 (Chen et al., 2023a) incorporate lightweight modules to extract features from the auxiliary modality while keeping the backbone of the foundation tracker frozen. SDSTrack (Hou et al., 2024) and OneTracker (Hong et al., 2024) simultaneously fine-tunes foundation tracker and a fusion module to extract modality-enhanced features. However, these approaches still demonstrate weaker performance than unified multi-modal foundation trackers trained from scratch, such as SUTrack (Chen et al., 2025) and FlexTrack (Tan et al., 2025). The second category of fine-tuning methods focus on strengthening the capabilities of the tracker itself, particularly spatio-temporal context modeling abilities. HIPTrack (Cai et al., 2024) introduces historical features by training historical prompt network. LoRAT (Lin et al., 2025) maintains a strong generalization by fine-tuning DINOv2 (Oquab et al., 2024). SPMTrack (Cai et al., 2025) further incorporates TMoE, auxiliary templates, and temporal propagation tokens for fine-tuning. Given the current landscape of unified multi-modal foundation trackers, the second category of PEFT methods offers better generality. Our method also falls into the second category.

### 2.2 TRACKERS WITH SPATIO-TEMPORAL CONTEXT.

Methods such as STARK (Yan et al., 2021a), MixFormer (Cui et al., 2022), and TrDiMP (Wang et al., 2021a) samples several historical frames as auxiliary templates. However, sparsely sampled frames overlook rich context and are prone to introduce distractors. Auxiliary templates also significantly increase the input sequence length, leading to an increase in computational overhead. Previous methods such as HIPTrack (Cai et al., 2024), to ensure computational efficiency, only fuse memory features with search region features after the backbone network, and additionally design a dedicated memory feature encoder. This results in more complex network architectures and insufficient fusion between memory features and search region features. In contrast, our MCP module compresses rich memory pool information into a fixed number of sparse memory tokens through elegant memory queries. This approach controls input sequence length and maintains computational efficiency while preserving rich memory information. Additionally, it enables deep fusion between memory tokens and search region features within the backbone network. Other methods like ODTrack (Zheng et al., 2024), SPMTrack (Cai et al., 2025) and AQATrack (Xie et al., 2024) introduce temporal propagated tokens across consecutive frames. MambaLT (Li et al., 2025), TemTrack (Xie et al., 2025), and STTrack (Hu et al., 2025) further collect temporal propagation tokens from multiple search regions and collectively enhance them using SSMs like Mamba (Gu & Dao, 2023). However, the limitation is that the propagation tokens attend to both the template and the search region, leading to the template diverting the attention of propagation tokens away from the search region and hinders the effective expression of target dynamic variations. The enhancement applied to these tokens also primarily serves to strengthen template features as well. Other methods like MambaVT (Lai et al., 2025) and MCITrack (Kang et al., 2025) utilize Mamba as an implementation of the trainable backbone, utilizing SSM's linear computational complexity to extend context length. However, these approaches require designing complex scanning algorithms, linear SSMs are also not guaranteed to outperform Transformers under the same sequence length (Merrill et al., 2024). In contrast, our DSF module serves a fundamentally different purpose. Unlike previous SSM methods that require designing complex scanning sequences and methodologies, DSF continuously

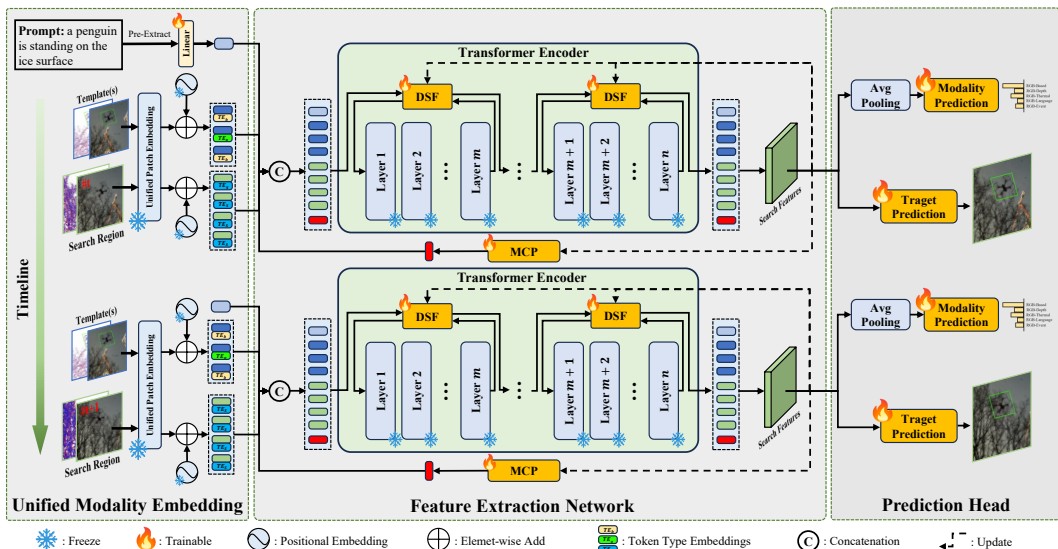

Figure 2: The Overall framework of Uni-MDTrack. Uni-MDTrack can uniformly process data from various modalities, and consists of unified modality embedding layer, feature extraction network and prediction head.

utilizes new frame search region features to update target states and employs these states as supplementary information for the backbone network. The key focus of the DSF module is to demonstrate that continuously updated dynamic target states can serve as efficient and effective complements to the backbone network, rather than being constrained to specific model designs. Furthermore, rather than re-architecting or replacing backbone layers, we inject memory and dynamic states with MCP and DSF in a lightweight adapter manner, eliminating the need of full-parameter training. To the best of our knowledge, we are the first to leverage SSM as a PEFT technique in the field of SOT.

## 3 METHOD

### 3.1 OVERALL ARCHITECTURE

As illustrated in Figure 2, we present **Uni-MDTrack**, a novel tracker built upon a prompt module based on memory-aware compression token (MCP) and a dynamic state fusion module based on SSMs (DSF), which supports tracking across five modalities: pure RGB, RGB-D, RGB-E, RGB-T, and RGB-Language. The overall architecture follows a one-stream paradigm, primarily consisting of a unified modality embedding layer, a feature extraction network based on HiViT (Zhang et al., 2023), and two prediction heads for target prediction and modality prediction. Throughout the model, the backbone remains frozen, with only MCP, DSF, and the prediction head serving as the main modules involved in training.

Specifically, input images from different modalities are first processed through the unified patch embedding module to generate a unified representation embedding. Positional embeddings and token type embeddings are then added to the unified representation embeddings. For text encoding in RGB-Language tracking tasks, we pre-extract the [cls] token as the text embedding using the pretrained text encoder from CLIP-L (Radford et al., 2021). All embedded tokens are fed into the feature extraction network. Within the feature extraction network, the memory-aware compression tokens output by MCP are first concatenated with the input tokens and input into the backbone. The backbone processes all tokens simultaneously, while performing fusion with dynamic state features via DSF from shallow to deep layers. Finally, we employ a center-based prediction head (Ye et al., 2022) to predict the tracking result , while a task recognition head (Chen et al., 2025) is used to predict the modality of the current input, thereby better assisting the model in capturing task-specific features.

## 3.2 UNIFIED MODALITY EMBEDDING

We adopt the same unified patch embedding module as (Chen et al., 2025). The unified patch embedding layer modifies the conventional patch embedding by extending the input dimension of linear projection layer from 3 to 6. RGB images are denoted as $\boldsymbol{X}_{\text{RGB}} \in \mathbb{R}^{H \times W \times 3}$, while depth, thermal, and event images are collectively referred to as DTE. We replicate DTE images into 3-channel images and normalize each pixel value to the range of [0, 255] to obtain $\boldsymbol{X}_{\text{DTE}} \in \mathbb{R}^{H \times W \times 3}$. We then construct a 6-channel input by concatenating the RGB and DTE image along the channel dimension to obtain $\boldsymbol{X} \in \mathbb{R}^{H \times W \times 6}$. For RGB and RGB-language tasks that do not include DTE images, we duplicate the RGB channels to form the required 6-channel input $\boldsymbol{X}$. Through the unified patch embedding process, template and search region images are transformed into respective token sequences $\boldsymbol{T} \in \mathbb{R}^{N_T \times d}$ and $\boldsymbol{S} \in \mathbb{R}^{N_S \times d}$. All tokens $\boldsymbol{T}$ and $\boldsymbol{S}$ are then added with positional embeddings soft token type embeddings following (Chen et al., 2025). For the RGB-Language tracking, the text feature token is projected through a small linear layer to obtain $\boldsymbol{L} \in \mathbb{R}^{1 \times d}$. All tokens are concatenated and fed into the feature extraction network.

## 3.3 FEATURE EXTRACTION NETWORK

Our feature extraction framework is built upon HiViT (Zhang et al., 2023). The feature extraction network first concatenates the input tokens with the memory-aware compression token $\boldsymbol{M} \in \mathbb{R}^{N_M \times d}$ output by the MCP module, resulting in token sequence $\boldsymbol{Z} \in \mathbb{R}^{N \times d}$ input to the backbone. The backbone network processes the sequence and continuously integrates dynamic state features from DSF modules. The final output sequence $\boldsymbol{O} \in \mathbb{R}^{N \times d}$ of the feature extraction network is used for the final target prediction. Meanwhile, output tokens corresponding to search region, denoted as $\boldsymbol{O_S}$, are added to the memory bank in MCP, as well as update the state in DSF modules.

### 3.3.1 MEMORY-AWARE COMPRESSION PROMPT MODULE (MCP)

The design of MCP is guided by two principles. First, introduce memory features at the input of the backbone, thereby enabling deep interaction with the template and search region features. Second, effectively compress memory features, thus maintaining a stable computational load for the overall model. Based on the above principles, we propose a dynamic query-based resampling approach for memory feature compression. Specifically, as shown in Figure 3(a), given the features $\boldsymbol{F}_m \in \mathbb{R}^{N_{mb} \times d}$ stored in the memory bank, MCP contains a total of $N_M$ trainable query tokens $\boldsymbol{q} \in \mathbb{R}^{N_M \times d}$, which perform dynamic querying and adaptive aggregation on $\boldsymbol{F}_m$ via these query tokens. The process can be formally described as follows:

$$\boldsymbol{Q} = \text{Linear}_q(\text{RMSNorm}_q(\boldsymbol{q}))$$
$$\boldsymbol{K}, \boldsymbol{V} = \text{Split}(\text{Linear}_{kv}(\text{RMSNorm}_{kv}(\boldsymbol{F}_m)))$$
$$\boldsymbol{Attn} = \text{Softmax}[\frac{\boldsymbol{Q} \cdot \boldsymbol{K}}{\sqrt{d}} + \text{ALiBi}(\boldsymbol{F}_m)] \tag{1}$$
$$\boldsymbol{M}_1 = \text{Linear}_o(\boldsymbol{Attn} \cdot \boldsymbol{V}) + \boldsymbol{q}$$
$$\boldsymbol{M}_2 = \text{FFN}(\text{RMSNorm}(\boldsymbol{M}_1)) + \boldsymbol{M}_1$$

where $\boldsymbol{Attn} \in \mathbb{R}^{N_M \times N_{mb}}$ is the attention weight, $\boldsymbol{M}_2 \in \mathbb{R}^{N_M \times d}$ is the memory features after query aggregation. For clarity, Equation 1 presents a simplified single-head attention formulation, although our actual implementation employs a multi-head mechanism. We also introduce an attention bias term $\text{ALiBi}(\boldsymbol{F}_m) \in \mathbb{R}^{N_{mb}}$ (Press et al., 2022), which provides $\boldsymbol{F}_m$ with extrapolatable positional information. ALiBi not only allows for significantly larger memory during inference but also effectively prioritizing recent memories over older, visually similar ones and mitigating the lingering effects of potential distractors. We encodes position at the frame level; assuming a token $\boldsymbol{F}_m^i$ originates from the $j^{th}$ frame in the memory bank, its corresponding bias is $-\boldsymbol{m}_h \times |j - N_{mb}|$, where $h$ is the index of attention head, and each head is associated with a unique slope $\boldsymbol{m}_h = 2^{\frac{-8}{h}}$. To prevent unbounded memory growth during inference, when the number of tracked frames exceeds $L$, we uniformly sample the search region tokens of $L$ frames from the tracked frames to serve as the memory bank. After obtaining the memory feature $\boldsymbol{M}_2$, we further use a self-attention module and a FFN layer to enhance it and output the final memory-aware compression tokens $\boldsymbol{M}$.

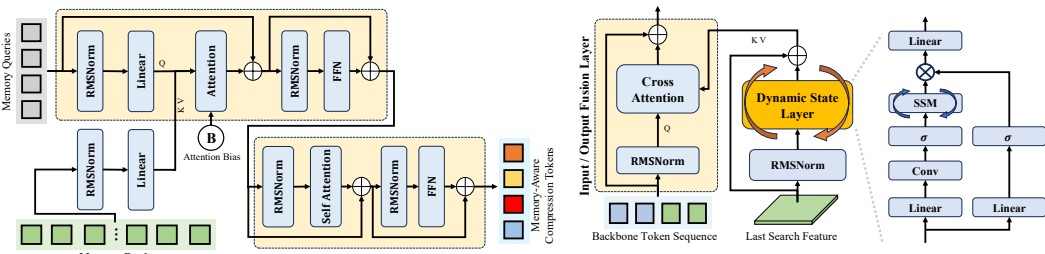

(a) The structure of Memory-Aware Compression Prompt Module (MCP)  (b) The structure of Dynamic State Fusion Module (DSF)

Figure 3: Detail structure of our proposed Memory-Aware Compression Prompt module (MCP) and Dynamic State Fusion module (DSF).

**Analysis of the Impact of ALiBi and Increasing Memory Length During Inference.** Let the current query be $q_t$ and the memory bank contain keys $(k_i)$ indexed by $i \in \mathcal{M}$. The attention scores are defined as: $a_{t,i} = \frac{q_t^\top k_i}{\sqrt{d}} + \beta \Delta(i,t)$ where $\Delta(i,t)$ is the relative distance, and $\beta < 0$ is ALiBi slope. Thus, the cross-attention logit and softmax weight are $p_{t,i} = \frac{e^{a_{t,i}}}{\sum_{j \in \mathcal{M}} e^{a_{t,j}}}$. Considering a simplified case where feature similarities are negligible ($q_t^\top k_i \approx q_t^\top k_j = 0$), the relative attention weight between two memories $i$ and $j$ depends solely on their distance: $\frac{p_{t,i}}{p_{t,j}} = \exp\left(\beta[\Delta(i,t) - \Delta(j,t)]\right)$. If memory $i$ is more recent than $j$ (i.e., $\Delta(i,t) < \Delta(j,t)$), then $p_{t,i} > p_{t,j}$. This provides a clean, explainable mechanism for prioritizing recent observations.

Assume the model is trained with a memory length $K$, but tested with length $L > K$. The total attention mass contributed by the unseen "tail" (memories beyond distance $K$) is bounded by a geometric series: $\text{Mass}_{\text{tail}} = \sum_{k=K+1}^{L} e^{\beta k} < \sum_{k=K+1}^{\infty} e^{\beta k} = \frac{e^{\beta(K+1)}}{1-e^\beta}$. To ensure the trained model's attention distribution remains valid during inference, we require this tail mass to be negligible (less than a threshold $\eta$). Solving $\frac{e^{\beta(K+1)}}{1-e^\beta} \leq \eta$ for $K$ yields: $K \gtrsim \frac{\ln(1/\eta)}{|\beta|} - 1$. This result implies that the effective memory horizon is of order $O(1/|\beta|)$. Consequently, extending the memory bank at test time adds only an exponentially small tail to the distribution, ensuring that our MCP module extrapolates robustly.

### 3.3.2 DYNAMIC STATE FUSION MODULE (DSF)

**Analysis of The Limitations of SSMs in Long-Sequence Extrapolation.** Formally, the hidden state update in Mamba is defined as: $H_t = \bar{A}_t \odot H_{t-1} + \bar{B}_t \odot X_t$, where $\bar{A}_t$ represents the channel-wise decay factors, and each element in $\bar{A}_t \in (0,1)$. A key structural constraint lies in the parameterization of $\bar{A}_t$: $\Delta_t = \text{Softplus}(X_t)$, $\bar{A}_t = \exp(\Delta_t \odot A)$, with $A < 0$ being a learnable matrix. Consequently, every dimension of $\bar{A}_t$ is strictly less than 1. By unrolling the recurrence, the contribution of an early token $X_j$ to the output at position $i$ is proportional to the cumulative product of decay factors: $\alpha_{i,j} \propto \left(\prod_{k=j+1}^{i} \bar{A}_k\right) \odot \bar{B}_j$. Leveraging $\bar{A}_k = \exp(\Delta_k \odot A)$, this product collapses into a unified exponential term: $\prod_{k=j+1}^{i} \bar{A}_k = \exp\left[\left(\sum_{k=j+1}^{i} \Delta_k\right) \odot A\right]$. Since $A < 0$ and $\Delta_k > 0$, there exists a constant $c > 0$ such that the magnitude of influence decays exponentially with distance: $\left\|\prod_{k=j+1}^{i} \bar{A}_k\right\| \leq \exp\left(-c(i-j)\right)$. This derivation leads to a direct conclusion: the influence of early tokens vanishes exponentially as the distance $i - j$ increases. While this decay may be manageable within the training length $L_{\text{train}}$, it becomes catastrophic during inference when extrapolating to $L_{\text{test}} \gg L_{\text{train}}$. Therefore, directly using an SSM as the backbone and simply extending the context length—as done in prior methods—will inevitably diminish the influence of earlier tokens. This is precisely why we employ an SSM-like structure only within DSF to model dynamic features, rather than introducing long-term memory.

DSF is conceived to meet two fundamental requirements: possessing enough capacity for capturing continuous target state dynamics, and enabling deep integration with the backbone. To achieve this, we deploy separate DSF modules at several hierarchical level of the backbone for state representation and updates. This multi-level approach provides a comprehensive view of the target dynamics while enabling deep integration with the backbone features. As shown in Figure 3(b), DSF module consists

of three key components: an input fusion layer, a dynamic state layer based on SSM, and an output fusion layer. The input fusion and output fusion layer respectively integrate the output target states feature $\boldsymbol{F}$ of dynamic state layer with the input $\boldsymbol{Z^i}$ of the $i^{th}$ backbone layer and output $\boldsymbol{O^j}$ of the $j^{th}$ layer, enabling us to freely configure both the quantity of DSF modules and the specific backbone layers with which they interact. The dynamic state layer utilizes only the search region features $\boldsymbol{O_S} \in \mathbb{R}^{N_S \times d}$ from the output of feature extraction network to perform state update, excluding the influence of the template or other tokens, thereby allowing the model to specifically capture the dynamics of the target itself. As shown in Figure 3(b), the overall process of dynamic state layer can be formally described as:

$$
\begin{aligned}
\boldsymbol{I} &= \text{RMSNorm}(\boldsymbol{O_S}) \\
\boldsymbol{G} &= \text{SiLU}(\text{Linear}_g(\boldsymbol{I})) \\
\boldsymbol{S_1} &= \text{SiLU}[\text{Conv}(\text{Linear}_c(\boldsymbol{I}))] \\
\boldsymbol{S} &= \text{SSM}(\boldsymbol{S_1}) \\
\boldsymbol{F} &= \boldsymbol{O_S} + \text{Linear}(\boldsymbol{G} \odot \boldsymbol{S})
\end{aligned}
\tag{2}
$$

The overall process follows a gated structure, where $\boldsymbol{G} \in \mathbb{R}^{N_S \times d_s}$ represents the gating values, and $d_s$ is the inner dimension of the dynamic state layer. $\boldsymbol{S_1} \in \mathbb{R}^{N_S \times d_s}$ is the input to the SSM after a linear projection and convolution-based activation, and $\boldsymbol{S} \in \mathbb{R}^{N_S \times d_s}$ is the output of the SSM. An element-wise multiplication of $\boldsymbol{G}$ with $\boldsymbol{S}$, followed by a linear layer, restores the dimensionality to $d$, yielding $\boldsymbol{F} \in \mathbb{R}^{N_S \times d}$. The state update and output of the SSM can be formally described as:

$$
\begin{aligned}
\boldsymbol{\Delta}, \boldsymbol{B}, \boldsymbol{C} &= \text{Split}(\text{Linear}(\boldsymbol{S_1})) \\
\bar{\boldsymbol{A}} &= \exp(\boldsymbol{\Delta A}), \\
\bar{\boldsymbol{B}} &= (\boldsymbol{\Delta A})^{-1}(\exp(\boldsymbol{\Delta A}) - \boldsymbol{I}) \cdot (\boldsymbol{\Delta B}) \\
h(t) &= \bar{\boldsymbol{A}}h(t-1) + \bar{\boldsymbol{B}}\boldsymbol{S}_1, \\
\boldsymbol{S} &= \boldsymbol{C}h(t) + \boldsymbol{D}\boldsymbol{S}_1
\end{aligned}
\tag{3}
$$

where $\boldsymbol{A}$ and $\boldsymbol{D}$ are learnable parameters, $\boldsymbol{I}$ is the identity matrix, and the entire process adheres to the discrete-time formulation of SSM. $h(t)$ denotes the hidden state at time $t$, with $h(0)$ initialized as a zero matrix. The SSM uses the current feature $\boldsymbol{S_1}$ to update $h(t)$ and generate the dynamic state $\boldsymbol{S}$ of the target. After obtaining $\boldsymbol{F}$ from the dynamic state layer, as shown in Figure 3(b), the input fusion layer is designed based on cross attention, integrates $\boldsymbol{F}$ with $\boldsymbol{Z^i}$ to produce the new input $\boldsymbol{Z^{i'}}$ of the $i^{th}$ backbone layer. The output fusion layer maintains an identical structure to the input fusion layer, integrating $\boldsymbol{F}$ with $\boldsymbol{O^j}$ to produce the new output $\boldsymbol{O^{j'}}$ of the $j^{th}$ backbone layer.

### 3.4 PREDICTION HEAD

The prediction head encompasses two tasks: target prediction and input modality classification. For target prediction, we first extract search region features $\boldsymbol{O_S}$ from the output $\boldsymbol{O}$ of the feature extraction network and feed $\boldsymbol{O_S}$ into the center-based prediction head (Ye et al., 2022). For input modality classification, we apply global average pooling to the output $\boldsymbol{O_S}$ and apply an MLP for classification.

## 4 EXPERIMENTS

### 4.1 IMPLEMENTATION DETAILS

**Model settings.**

We propose two versions of our tracker, Uni-MDTrack-B and Uni-MDTrack-L. Uni-MDTrack-B employs a template size of $112 \times 112$ and a search region size of $224 \times 224$, while Uni-MDTrack-L uses $196 \times 196$ and $384 \times 384$, respectively. The cropping factors for the template and search region are 2.0 and 4.0 for both versions. Uni-MDTrack-B and Uni-MDTrack-L adopt HiViT-B and HiViT-L (Zhang et al., 2023) as backbones, respectively, and are initialized with the same weights as SUTrack-B and SUTrack-L (Chen et al., 2025). MCP module outputs a total of 16 memory-aware compression tokens, and the number of attention heads is kept consistent with backbone. We employ a total of four DSF modules. For Uni-MDTrack-B, we divide its last 24 backbone layers into four

equal segments, and for Uni-MDTrack-L, we do the same for its last 40 layers. The dynamic state features are then fused with the input and output of each of these four segments.

Table 1 details the parameter and computational overhead of our models. Compared with other methods that use PEFT to enhance model capabilities, our method has a significant advantage in computational cost while introducing a comparable number of additional training parameters.

Table 1: Comparison of our method with other trackers using parameter-efficient training method in terms of total parameters, trainable parameters, and computational complexity.

| Method | Trainable Params(M) | Params(M) | FLOPs(G) |
|---|---|---|---|
| **Uni-MDTrack-B** | 27.1 | **88.2** | **27.9** |
| HIPTrack (Cai et al., 2024) | 34.1 | 120.4 | 66.9 |
| LoRAT-B$_{384}$ (Lin et al., 2025) | **13.0** | 99.1 | 97.0 |
| SPMTrack-B (Cai et al., 2025) | 29.2 | 115.3 | - |
| **Uni-MDTrack-L** | 54.9 | 287.4 | 257.4 |

**Datasets.** Following SUTrack (Chen et al., 2025), our method utilizes the *training* sets from LaSOT (Fan et al., 2019), GOT-10K (Huang et al., 2019), COCO (Lin et al., 2014), TrackingNet (Muller et al., 2018), VastTrack (Peng et al., 2024b), TNL2K (Wang et al., 2021b), DepthTrack (Yan et al., 2021b), VisEvent (Wang et al., 2024), and LasHeR (Li et al., 2022) for training. During training, in each batch sampling step, the probability ratios used for sampling from each dataset are set to 2:2:2:2:2:2:1:1:1. We sample 7 frames per step, with the first 2 frames serving as templates and the latter 5 as search frames. For the image dataset COCO, we replicate single images multiple times to simulate sequential data.

**Training and Optimization.** Our method is implemented based on PyTorch 2.3.1 and trained on 4 NVIDIA A100 GPUs. We set the batch size to 64 per GPU for Uni-MDTrack-B and 16 for Uni-MDTrack-L. Both version are trained for 50 epochs, with 100,000 frame sequences sampled from all datasets in each epoch. We employ the AdamW (Loshchilov & Hutter, 2019) optimizer with an initial learning rate of 2e-4 for both stages, which is decreased to 2e-5 after 40 epochs. The weight decay is set to 1e-4 throughout the training process.

**Loss Function.** For target prediction, consistent with OSTrack (Ye et al., 2022), we employ Generalized IoU (Rezatofighi et al., 2019) Loss and L1 Loss to supervise bounding box prediction, and Focal Loss (Lin et al., 2017) to supervise target center point prediction. Additionally, we use Cross-Entropy Loss to compute the modality prediction loss. The loss weights for the above components are set to 2.0, 5.0, 1.0, and 1.0, respectively.

**Inference.** Consistent with SUTrack (Chen et al., 2025), we use 2 templates input. DSF module performs continuous state updates per frame during tracking, and the memory bank of MCP contains a total of 50 frames of uniformly sampled historical search region features.

## 4.2 COMPARISONS WITH THE STATE-OF-THE-ART METHODS

Table 2: State-of-the-art comparison on RGB-T tracking dataset LasHeR, RGB-E tracking dataset VisEvent, and RGB-D tracking dataset DepthTrack. The best three results are highlighted in red, blue and **bold**, respectively.

| Method | Source | LasHeR | | VisEvent | | DepthTrack | | |
|---|---|---|---|---|---|---|---|---|
| | | SR(%) | PR(%) | AUC(%) | P(%) | F-Score(%) | Re(%) | PR(%) |
| **Uni-MDTrack-B** | **Ours** | 61.2 | 76.7 | 64.2 | **81.0** | 65.9 | 66.3 | 66.2 |
| **Uni-MDTrack-L** | **Ours** | 62.1 | 77.9 | 65.7 | 81.8 | 67.4 | 67.2 | 67.6 |
| FlexTrack (Tan et al., 2025) | ICCV25 | 62.0 | 77.3 | 64.1 | 81.4 | 67.0 | 66.9 | 67.1 |
| SUTrack-B$_{224}$ (Chen et al., 2025) | AAAI25 | 59.9 | 74.5 | 62.7 | 79.9 | 65.1 | 65.7 | 64.5 |
| SUTrack-L$_{384}$ (Chen et al., 2025) | AAAI25 | **61.9** | **76.9** | 63.8 | 80.5 | **66.4** | **66.4** | **66.5** |
| STTrack (Hu et al., 2025) | AAAI25 | 60.3 | 76.0 | 61.9 | 78.6 | 63.3 | 63.4 | 63.2 |
| SeqTrackV2-B$_{256}$ (Chen et al., 2023a) | Arxiv23 | 55.8 | 70.4 | 61.2 | 78.2 | 63.2 | 63.4 | 62.9 |
| UnTrack (Wu et al., 2024) | CVPR24 | 53.6 | 66.7 | 58.9 | 75.5 | 61.2 | 61.0 | 61.3 |
| SDSTrack (Hou et al., 2024) | CVPR24 | 53.1 | 66.5 | 59.7 | 76.7 | 61.4 | 60.9 | 61.9 |
| OneTracker (Hong et al., 2024) | CVPR24 | 53.8 | 67.2 | 60.8 | 76.7 | 60.9 | 60.4 | 60.7 |
| ViPT (Zhu et al., 2023) | CVPR23 | 52.5 | 65.1 | 59.2 | 75.8 | 59.4 | 59.6 | 59.2 |

To better compare with mainstream RGB trackers, we additionally trained a pure RGB tracker MDTrack-B, which is implemented based on SPMTrack-B (Cai et al., 2025). The prediction head of MDTrack-B no longer introduces modality prediction. Similarly, we divided all 12 layers of the model into 4 equal parts to fuse DSF modules, and trained it only on LaSOT, GOT-10K, TrackingNet, and COCO with identical training configurations.

Table 3: State-of-the-art comparison on RGB visual tracking datasets LaSOT, TrackingNet and LaSOT$_{ext}$. The best three results are highlighted in **red**, **blue** and **bold**, respectively.

| Method | Source | LaSOT | | | TrackingNet | | | LaSOT$_{ext}$ | | |
|---|---|---|---|---|---|---|---|---|---|---|
| | | AUC(%) | $P_{Norm}$(%) | $P$(%) | AUC(%) | $P_{Norm}$(%) | $P$(%) | AUC(%) | $P_{Norm}$(%) | $P$(%) |
| *Unified Trackers* | | | | | | | | | | |
| **Uni-MDTrack-B** | **Ours** | 74.7 | 84.9 | 82.6 | 86.1 | 90.8 | 85.9 | 54.3 | 65.7 | 62.4 |
| **Uni-MDTrack-L** | **Ours** | 76.1 | 85.7 | 84.3 | 88.0 | 92.1 | 89.1 | 55.2 | 66.3 | 62.8 |
| SUTrack-B$_{224}$ (Chen et al., 2025) | AAAI25 | 73.2 | 83.4 | 80.5 | 85.7 | 90.3 | 85.1 | 53.1 | 64.2 | 60.5 |
| SUTrack-L$_{384}$ (*Chenet al.*, 2025) | AAAI25 | 75.2 | 84.9 | 83.2 | 87.7 | 91.7 | 88.7 | 53.6 | 64.2 | 60.5 |
| *RGB-based Trackers* | | | | | | | | | | |
| **MDTrack-B** | **Ours** | 75.6 | 85.1 | 83.8 | 86.4 | 90.6 | 86.2 | 54.8 | 65.6 | 62.1 |
| SPMTrack-B (Cai et al., 2025) | CVPR25 | 74.9 | 84.0 | 81.7 | 86.1 | 90.2 | 85.6 | - | - | - |
| ARPTrack$_{256}$ (Liang et al., 2025) | CVPR25 | 72.6 | 81.4 | 78.5 | 85.5 | 90.0 | 85.3 | 52.0 | 62.9 | 58.7 |
| MCITrack-B (Kang et al., 2025) | AAAI25 | 75.3 | 85.6 | 83.3 | 86.3 | 90.9 | 86.1 | 54.6 | 65.7 | 62.1 |
| MambaLCT$_{384}$ (Li et al., 2025) | AAAI25 | 73.6 | 84.1 | 81.6 | 85.2 | 89.8 | 85.2 | 53.3 | 64.8 | 61.4 |
| LoRAT-B$_{378}$ (Lin et al., 2025) | ECCV24 | 72.9 | 81.9 | 79.1 | 84.2 | 88.4 | 83.0 | 53.1 | 64.8 | 60.6 |
| AQATrack$_{384}$ (Xie et al., 2024) | CVPR24 | 72.7 | 82.9 | 80.2 | 84.8 | 89.3 | 84.3 | 52.7 | 64.2 | 60.8 |
| ARTrackV2-B$_{384}$ (Bai et al., 2024) | CVPR24 | 73.0 | 82.0 | 79.6 | 85.7 | 89.8 | 85.5 | 52.9 | 63.4 | 59.1 |
| HIPTrack (Cai et al., 2024) | CVPR24 | 72.7 | 82.9 | 79.5 | 84.5 | 89.1 | 83.8 | 53.0 | 64.3 | 60.6 |
| ODTrack-B (Zheng et al., 2024) | AAAI24 | 73.2 | 83.2 | 80.6 | 85.1 | 90.1 | 84.9 | 52.4 | 63.9 | 60.1 |
| ARTrack$_{384}$ (Wei et al., 2023) | CVPR23 | 72.6 | 81.7 | 79.1 | 85.1 | 89.1 | 84.8 | 51.9 | 62.0 | 58.5 |
| SeqTrack-B$_{384}$ (Chen et al., 2023b) | CVPR23 | 71.5 | 81.1 | 77.8 | 83.9 | 88.8 | 83.6 | 50.5 | 61.6 | 57.5 |
| OSTrack$_{384}$ (Ye et al., 2022) | ECCV22 | 71.1 | 81.1 | 77.6 | 83.9 | 88.5 | 83.2 | 50.5 | 61.3 | 57.6 |

**LaSOT** (Fan et al., 2019) is an *RGB-based* tracking dataset constructed for long-term tracking. As shown in Table 3, our approach achieves significant improvement compared to SUTrack-B$_{224}$ (**+1.5 AUC**) and SUTrack-L$_{384}$ (**+0.9 AUC**). MDTrack-B also achieves significant improvement compared to SPMTrack-B (**+0.7 AUC**) and outperforms all RGB Trackers based on ViT-B (Dosovitskiy et al., 2021).

**LaSOT$_{ext}$** (Fan et al., 2019) is an *RGB-based* tracking dataset that has no overlaps with LaSOT (Fan et al., 2021). As shown in Table 3, our method significantly outperforms SUTrack and MDTrack-B achieves the best performance among RGB trackers.

**TrackingNet** (Muller et al., 2018) is an *RGB-based* large-scale tracking dataset. As shown in Table 3, our method outperforms SUTrack (Chen et al., 2025), and MDTrack-B outperforms other RGB trackers.

**UAV123, OTB2015 and NfS** (Mueller et al., 2016; Wu et al., 2015; Kiani Galoogahi et al., 2017) are *RGB-based* datasets. We conduct evaluations on the 30 FPS version of NfS. As shown in Table 5, our method significantly outperforms SUTrack-B$_{384}$ with a larger resolution.

**TNL2K** (Wang et al., 2021b) is an *RGB-Language* tracking dataset. Each video is accompanied by natural language description. As shown in Table 4, our method also significantly outperforms SUTrack-B$_{224}$ (**+2.6 AUC**) and SUTrack-L$_{384}$ (**+2.5 AUC**). Our approach outperforming existing state-of-the-art methods by a significant gap.

Table 4: The performance of our method and other state-of-the-art trackers on RGB-Language Tracking dataset TNL2K. The best three results are highlighted in **red**, **blue** and **bold**.

| Method | AUC(%) | $P_{Norm}$(%) | $P$(%) |
|---|---|---|---|
| **Uni-MDTrack-B** | 67.6 | 85.2 | 73.2 |
| **Uni-MDTrack-L** | 70.4 | 87.4 | 77.4 |
| SUTrack-B$_{224}$ (Chen et al., 2025) | 65.0 | - | 67.9 |
| SUTrack-L$_{384}$ (Chen et al., 2025) | 67.9 | - | 72.1 |
| MCITrack-B (Kang et al., 2025) | 62.9 | - | - |
| LoRAT-B$_{378}$ (Lin et al., 2025) | 59.9 | - | 63.7 |
| ODTrack-L (Zheng et al., 2024) | 61.7 | - | - |
| ARTrackV2-L$_{384}$ (Bai et al., 2024) | 61.6 | - | - |
| CiteTracker (Li et al., 2023) | 57.7 | - | 59.6 |
| VLT (Guo et al., 2022) | 53.1 | - | 53.3 |

Table 5: The performance of our method and other state-of-the-art trackers on UAV123, NfS and OTB2015 in terms of AUC metrics. The best three results are highlighted in **red**, **blue** and **bold**.

| Method | UAV123 | NfS | OTB2015 |
|---|---|---|---|
| **Uni-MDTrack-B** | 71.0 | 70.2 | 73.6 |
| SUTrack-B$_{384}$ (Chen et al., 2025) | 70.4 | 69.3 | - |
| HIPTrack (Cai et al., 2024) | 70.5 | 68.1 | 71.0 |
| ARTrackV2-B (Bai et al., 2024) | 69.9 | 67.6 | - |
| ODTrack-L (Zheng et al., 2024) | - | - | 72.4 |
| ARTrack$_{384}$ (Wei et al., 2023) | 70.5 | 66.8 | - |
| SeqTrack-B$_{384}$ (Chen et al., 2023b) | 68.6 | 66.7 | - |
| MixFormer-L (Cui et al., 2022) | 69.5 | - | - |

**DepthTrack** (Yan et al., 2021b) is an *RGB-Depth* tracking dataset. As shown in Table 2, our method achieves state-of-the-art performance on DepthTrack.

**LasHeR** (Li et al., 2022) is an *RGB-Thermal* short-term tracking dataset with high-diversity. As shown in Table 2, our method achieves current best performance and shows a remarkable boost in terms of PR.

**VisEvent** (Wang et al., 2024) is an *RGB-Event* tracking dataset. As shown in Table 2, our method achieves state-of-the-art performance, demonstrating significantly higher performance than SU-Track (Chen et al., 2025). Uni-MDTrack-B can achieve even better results than SUTrack-L$_{384}$.

## 4.3 ABLATION STUDY

**The Importance of MCP and DSF.** In Table 6, based on Uni-MDTrack-B, we conduct ablation studies on the two core components of our method: MCP and DSF. Results demonstrates that both our proposed MCP and DSF modules individually contribute to significant tracking performance improvements. MCP proves more critical for long-term (LaSOT) and infrared (LasHeR) tracking, whereas DSF generally has a greater impact on the remaining datasets.

**Comparison with Other PEFT Methods.** Table 7 presents a comparative analysis of our method against two prominent PEFT methods: HIPTrack and SPMTrack. To ensure a fair and controlled comparison, all methods are trained on the same foundation model DropTrack (Wu et al., 2023), which is a variant of the well-established OSTrack (Ye et al., 2022), featuring an enhanced initialization strategy. Our method achieved a more significant performance improvement with only 50 epochs of training.

Table 6: Ablation studies on MCP and DSF modules. Experiments are conducted on LaSOT (evaluated by AUC), LasHeR (SR), VisEvent (AUC) and DepthTrack (F-Score).

| # | MCP | DSF | LaSOT | LasHeR | VisEvent | DepthTrack | Δ |
|---|-----|-----|-------|--------|----------|------------|-------|
| 1 | ✗ | ✗ | 73.2 | 59.9 | 62.7 | 65.1 | 0 |
| 2 | ✗ | ✔ | 73.8 | 60.4 | 63.6 | 65.7 | +0.65 |
| 3 | ✔ | ✗ | 74.1 | 60.6 | 63.3 | 65.3 | +0.6 |
| 4 | ✔ | ✔ | 74.7 | 61.2 | 64.2 | 65.9 | +1.3 |

Table 7: A performance comparison of existing trackers and their integration with our method on LaSOT *test* set.

| Method | AUC(%) | $P_{Norm}$(%) | $P$(%) |
|--------|--------|---------------|--------|
| DropTrack | 71.8 | 81.8 | 78.1 |
| DropTrack *w/* **Ours** | **73.1** | 82.7 | **79.7** |
| DropTrack *w/* HIP | 72.7 | **82.9** | 79.5 |
| DropTrack *w/* Temporal Token | 72.0 | 81.9 | 78.4 |

**Generalization Ability of Our Method.** The results across Tables 2, 3, and 7 collectively demonstrate the remarkable generalization ability and effectiveness of our proposed MCP and DSF modules. Our method consistently delivers substantial performance gains when applied to three distinct excellent trackers, proving performance gain in both pure RGB and unified multi-modal tracking scenarios.

Table 8: Ablation study on different number of memory-aware compression tokens.

| Number | 8 | 16 | 32 | 64 |
|--------|-----|-----|-----|-----|
| LaSOT | 74.2 | 74.7 | 74.7 | 74.6 |
| LasHeR | 60.5 | 61.2 | 61.3 | 61.3 |
| VisEvent | 63.8 | 64.2 | 64.2 | 64.3 |
| DepthTrack | 65.5 | 65.9 | 65.9 | 66.1 |
| Δ | -0.5 | 0 | +0.03 | +0.08 |

**The Number of Memory-Aware Compression Token.** Table 8 explores the impact of the number of memory-aware compression tokens based on Uni-MDTrack-B. While performance increases with more tokens, it eventually saturates. We thus selected 16 as a balanced choice.

## 5 CONCLUSION

This paper presents a simple, efficient PEFT method for single object tracking, centered on two modules: Memory-Aware Compression Prompt module (MCP) and Dynamic State Fusion module (DSF). These modules achieve a deep fusion of memory features and continuous dynamic state of the target, enhancing tracking performance while preserving efficiency. Based on the MCP and DSF modules, we design Uni-MDTrack, which supports five modalities and achieves new state-of-the-art performance as an unified tracker by training 30% of its parameters. Crucially, both MCP and DSF demonstrate strong generalizability, functioning as effective plug-and-play enhancements for various trackers. We hope this work encourages more low-cost, high-efficiency research in single object tracking.

## ETHICS STATEMENT

Our work adheres to the code of ethics.

## REPRODUCIBILITY STATEMENT

In Section 3 of our paper, we provide a detailed description of the overall method pipeline and the design of each module. Meanwhile, we use formal formulas to describe each operation. Additionally, in the Experiment Section 4, we elaborate on the hyperparameter configurations of the experiments, the baselines used, and the model parameter count as well as computational cost. All these description also serve as a reference for reproduction. The datasets we used are all public datasets, which also provides a guarantee for reproduction. We will make the source code and trained model weights publicly available after the paper is accepted.

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

## USAGE OF LLM

During the writing process, we use LLMs only to assist in checking English spelling and grammar, as well as to standardize academic writing.

## APPENDIX

In the supplementary material, Section A further reports the experimental results of our method, including results on other datasets and more ablation experiments, to further verify the effectiveness of Uni-MDTrack and our proposed method. Section B provides more detailed success curves on the overall LaSOT dataset and its subsets, as well as the overall precision curve. Section C demonstrates the visualization performance comparisons of various trackers in complex scenarios, while presenting more qualitative analyses of our method.

## A    A. MORE RESULTS AND FURTHER ANALYSIS

### A.1    PERFORMANCE ON GOT-10K

The reason we do not include the test results for GOT-10k (Huang et al., 2019) in the main paper is that previous trackers, when tested on GOT-10k, are typically trained only on the GOT-10k dataset itself (Cai et al., 2025; Bai et al., 2024; Wei et al., 2023; Ye et al., 2022; Fu et al., 2022). Since our proposed Uni-MDTrack is a tracker that supports all modalities and is trained on datasets from multiple modalities, a direct comparison with previous methods on GOT-10k would not be fair. Therefore, we have provided the results of our method in Table A of the supplementary material for reference.

Table A: The performance of our method and other state-of-the-art trackers on RGB-based Tracking dataset GOT-10k. The best three results are highlighted in **red**, **blue** and **bold**.

| Method | AO(%) | $SR_{0.5}$(%) | $SR_{0.75}$(%) |
|---|---|---|---|
| **Uni-MDTrack-B** | **81.1** | **91.8** | **81.2** |
| SUTrack-B$_{224}$ (Chen et al., 2025) | **77.9** | **87.5** | **78.5** |
| SPMTrack-B (Cai et al., 2025) | 76.5 | 85.9 | 76.3 |
| MCITrack-B (Kang et al., 2025) | **77.9** | **88.2** | **76.8** |
| ARPTrack$_{256}$ (Liang et al., 2025) | 77.7 | 87.3 | 74.3 |
| MambaLCT$_{384}$ (Li et al., 2025) | 76.2 | 86.7 | 74.3 |
| LoRAT-B$_{378}$ (Lin et al., 2025) | 73.7 | 82.6 | 72.9 |

### A.2    MORE ABLATION STUDY ON DSF

**The Number of DSF Modules.** Our standard model employs four DSF modules, with each DSF module uniformly fusing target dynamic state features into the input and output of certain layers in the backbone network. In Table B, we attempted to introduce different numbers of DSF modules based on Uni-MDTrack-B, respectively. We still divided its last 24 layers into equal parts. The results show that using 4 DSF modules achieves the best performance; more DSF modules will introduce additional parameters, and 50 epochs of training may not be sufficient to fully train these modules. Meanwhile, due to time constraints, we do not conduct further experiments on Uni-MDTrack-L, so we adopt the same settings as Uni-MDTrack-B.

Table B: Ablation study on different number of DSF modules based on Uni-MDTrack-B.

| Number | 2 | 4 | 6 | 8 |
|---|---|---|---|---|
| LaSOT | 74.3 | 74.7 | 74.7 | 74.6 |
| LasHeR | 61.0 | 61.2 | 61.4 | 61.2 |
| VisEvent | 63.6 | 64.2 | 64.2 | 64.0 |
| DepthTrack | 65.4 | 65.9 | 65.9 | 65.7 |
| $\Delta$ | -0.43 | 0 | +0.05 | -0.13 |

**Which Features to Use for Target State Updates.** A core design principle of our method is to update the state of all DSF modules using only the final search region features. Isolating the state

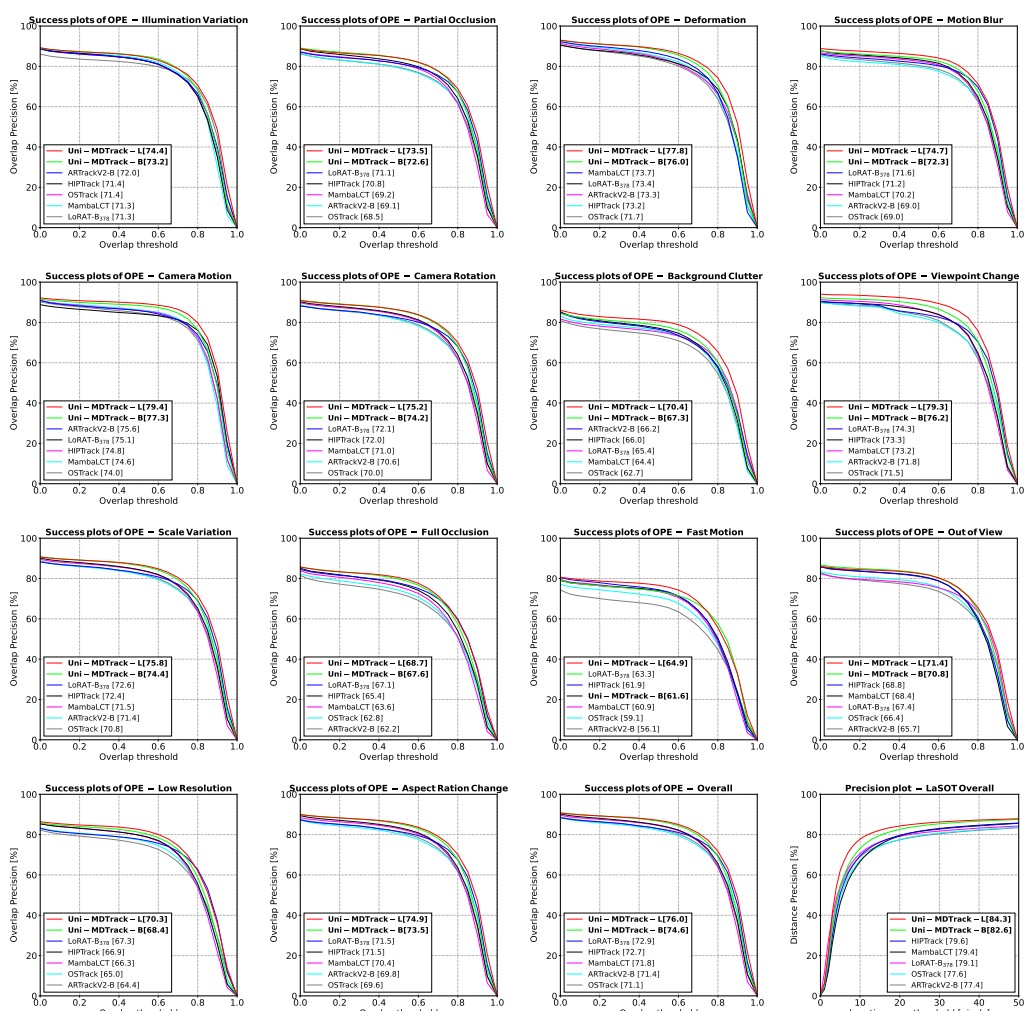

Figure A: Comparisons of our proposed Uni-MDTrack with other excellent trackers in the success curve on LaSOT *test* split, which includes eleven challenging scenarios such as Low Resolution, Motion Blur, Scale Variation, etc. We also provide the comparisons of the success and precision curves across the entire LaSOT *test* split. Zoom in for better view.

update from template information ensures that the DSF modules are solely dedicated to capturing the real-time dynamics of the target. Table C presents an ablation study where we validate this design choice by investigating the impact of using different feature sources for the state update. The second row of Table C represents using the overall output sequence to perform SSM state updates; the third row represents using the search region features corresponding to the input of the DSF module's input fusion layer to perform updates, *i.e.*, the input for state updates of each DSF module comes from different intermediate layers of the backbone. The results show that introducing other features reduces model performance; meanwhile, using intermediate network layer features to update the target state is not as effective as using the final search region features.

Table C: Ablation studies on which features to use for target state updates in DSF.

| # | Variants | LaSOT | LasHeR | VisEvent | DepthTrack | Δ |
|---|----------|-------|--------|----------|------------|---|
| 1 | Output Search Feature | 74.7 | 61.2 | 64.2 | 65.9 | 0 |
| 2 | Output Whole Sequence | 74.5 | 60.9 | 63.9 | 65.7 | **-0.25** |
| 3 | Input Fusion Sequence | 74.5 | 60.7 | 64.1 | 65.6 | **-0.28** |

**The Impact of Using State Space Model.** Our DSF module employs a Mamba-like SSM (Gu & Dao, 2023). In fact, in addition to state space models, there are many RNN-like structures that can achieve state updates. As shown in Table D, we also experimented with replacing SSM with a

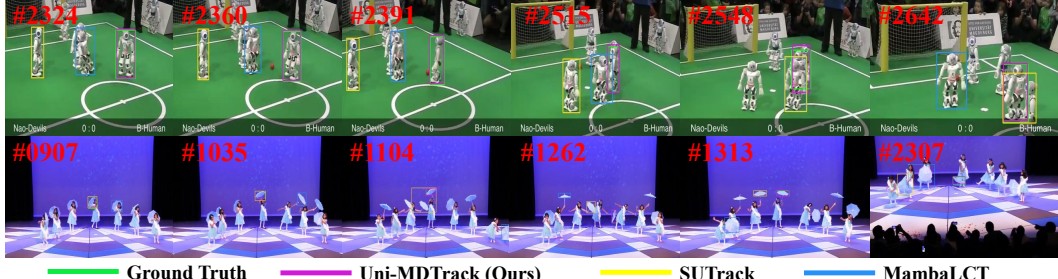

(a) Qualitative results of three methods when the targets in large scale variations.

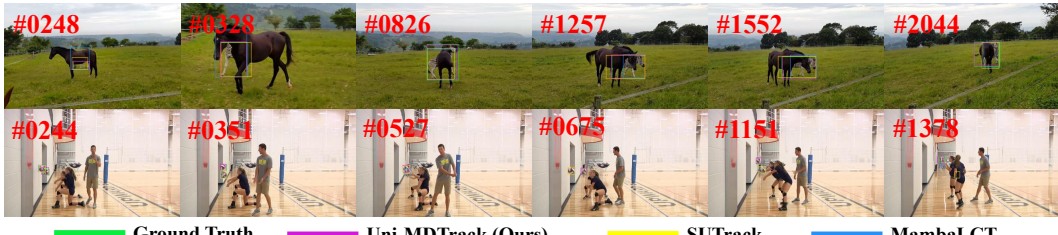

(b) Qualitative results of three methods when the targets are among similar objects and suffer partial occlusion.

(c) Qualitative results of three methods when the targets have large occlusions and sudden moves.

Figure B: This figure presents a visual comparison among our proposed Uni-MDTrack, MambaLCT$_{256}$ (Li et al., 2025) and SUTrack-B (Chen et al., 2025) in the challenges of target among similar objects, undergoes sudden movements, partial occlusion and scale variation. It demonstrates that our method achieves more effective and accurate tracking in the aforementioned challenging scenarios. Zoom in for better view.

simple LSTM. The results show that SSM can achieve better performance due to better state update algorithms, but LSTM is still effective. We also believe that replacing SSM with modern recurrent units such as RWKV (Peng et al., 2024a) can achieve better performance.

**Comparison with Using SSMs as Backbone Layers.** Previous methods such as MambaVT (Lai et al., 2025) and MambaVLT (Liu et al., 2025) predominantly use SSM as the backbone network implementation, or replace certain layers of the backbone with SSM, fundamentally playing the same role as other backbone layers while merely utilizing SSM's linear computational complexity to extend context length. These approaches also require designing complex scanning algorithms. But linear SSMs are not guaranteed to outperform Transformers under the same sequence length (as proven in (Merrill et al., 2024)). We have conducted experiments demonstrating this limitation, where replacing our DSF module with the same number of encoder layers from MambaVT and inserting them at regular intervals throughout the backbone network, which results in substantial performance degradation, as shown in Table E.

Table D: Ablation studies on replacing SSM with other model.

| # | Variants | LaSOT | LasHeR | VisEvent | DepthTrack | Δ |
|---|----------|-------|--------|----------|------------|------|
| 1 | SSM | 74.7 | 61.2 | 64.2 | 65.9 | 0 |
| 2 | LSTM | 74.4 | 60.8 | 63.8 | 65.7 | **-0.33** |

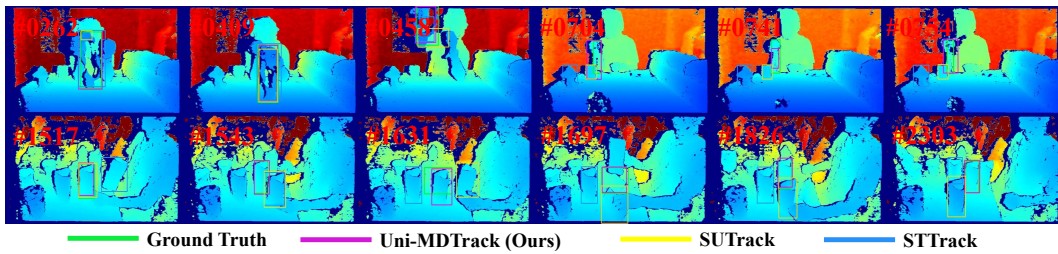

(a) Qualitative results of three methods on RGB-Depth tasks.

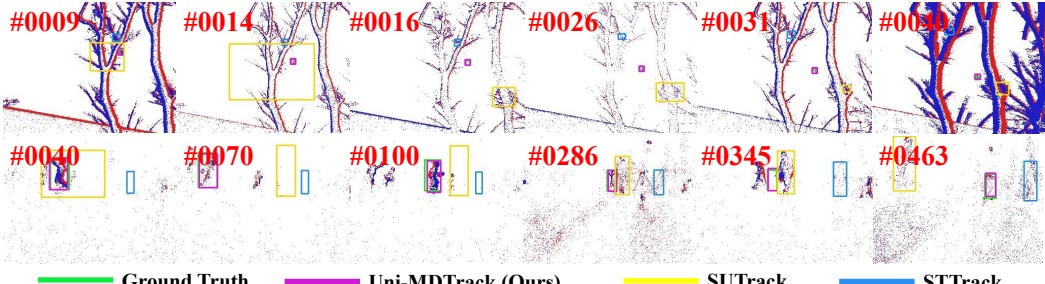

(b) Qualitative results of three methods on RGB-Event tasks.

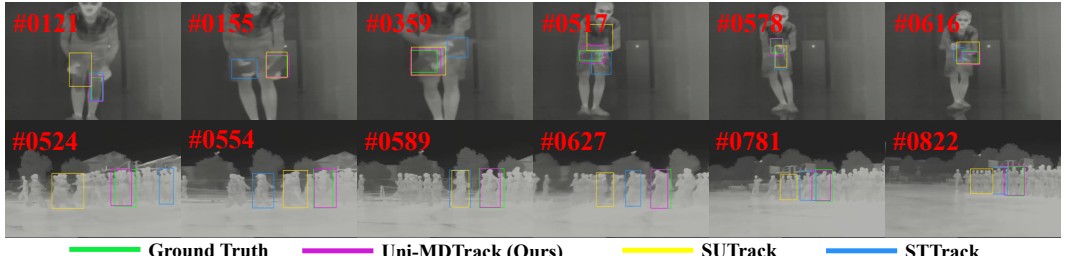

(c) Qualitative results of three methods on RGB-Thermal tasks.

Figure C: This figure presents a visual comparison among our proposed Uni-MDTrack-B, STTrack (Hu et al., 2025) and SUTrack-B (Chen et al., 2025) in the challenges of different modalities including RGB-Depth, RGB-Event and RGB-Thermal tasks. It demonstrates that our method achieves more effective and accurate tracking in the aforementioned challenging scenarios. Zoom in for better view.

Table E: Ablation studies on leveraging SSMs as backbone layers.

| # | Variants | LaSOT | LasHeR | VisEvent | DepthTrack | Δ |
|---|---|---|---|---|---|---|
| 1 | Ours | 74.7 | 61.2 | 64.2 | 65.9 | 0 |
| 2 | Backbone Layers | 74.0 | 60.5 | 63.5 | 65.4 | -0.65 |

### A.3 MORE ABLATION STUDY ON MCP

**The Size of Memory Bank.** In this paper, we construct the memory bank by sampling the search region features from $n$ tracked frames, which are selected uniformly and at equal intervals from all previously tracked frames. By default, we set the number $n$ to 50. In Table F, we further experiment with different memory bank sizes. When the memory bank size is further increased, model performance ceases to improve; thus, we select 50 to conserve GPU memory. This may be due to the limited number of queries and network layers of MCP, which restrict the compression capability for large memory bank. Additionally, since DSF can capture the short-term dynamic state changes of the target, MCP is not required to sample more densely.

**The Impact of Position Bias Added to Attention Score.** When using queries for memory feature querying and aggregation, we incorporate the ALiBi (Press et al., 2022) positional bias for different memory frames to maintain the positional awareness of memory frames, enabling the model to pay

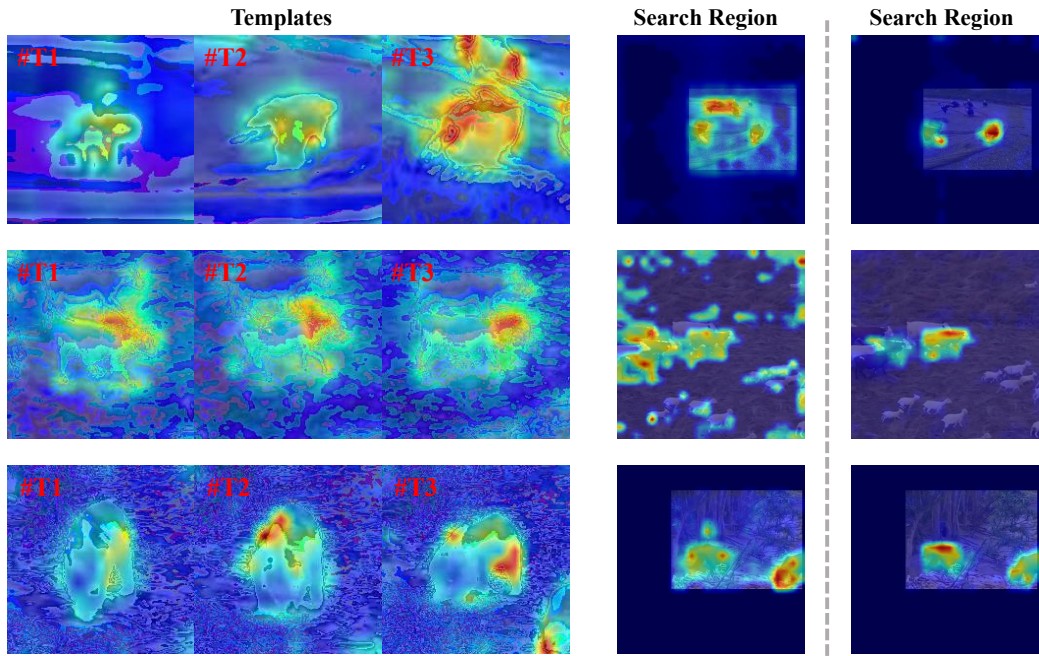

Attention map of the temporal propagating token on the Templates and Search Region in ODTrack          Ours

Figure D: Visualization of the attention of the Temporal Propagate Token on the template and search region in ODTrack, compared with the visualization of the attention between the search region and the Dynamic State Feature in DSF module of our method.

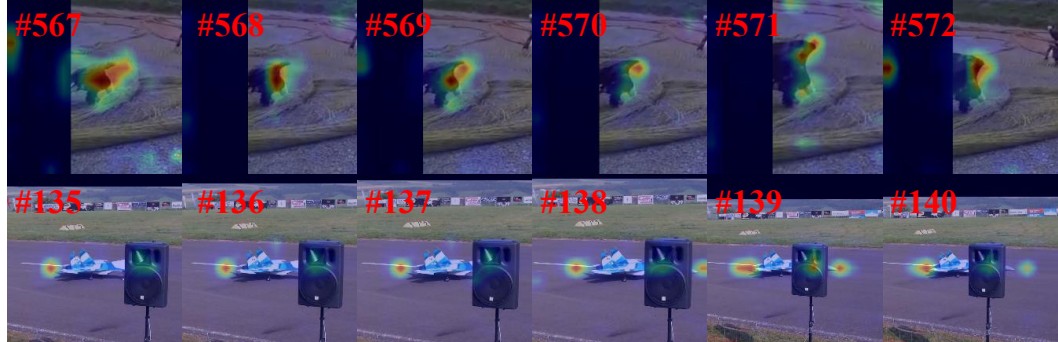

Figure E: Visualization of the attention map between the search region and the Dynamic State Feature output by DSF module when the target undergoes scale variation and fast motion.

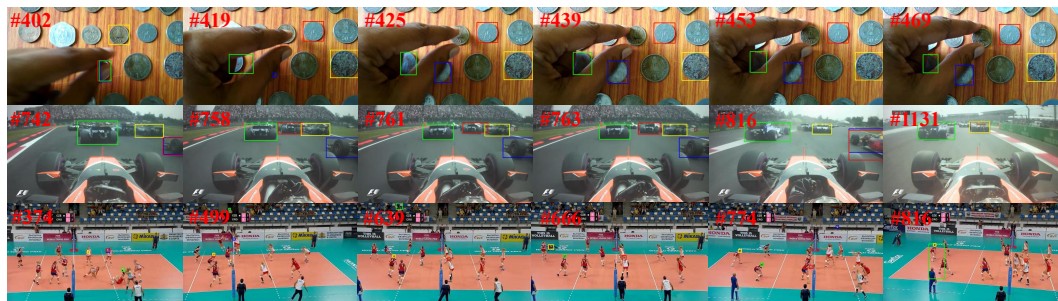

Figure F: This figure illustrates cases of tracking failure under extremely complex scenarios. When the target undergoes severe occlusion and fast motion, while heavy background clutter are present, all methods experience tracking failures. Zoom in for better view.

Table F: Ablation study on memory bank size of MCP module based on Uni-MDTrack-B.

| Number | 10 | 20 | 50 | 100 |
|---|---|---|---|---|
| LaSOT | 74.3 | 74.6 | 74.7 | 74.6 |
| LasHeR | 60.7 | 61.0 | 61.2 | 61.0 |
| VisEvent | 63.9 | 64.0 | 64.2 | 64.4 |
| DepthTrack | 65.9 | 66.0 | 65.9 | 65.8 |
| $\Delta$ | -0.3 | -0.1 | 0 | -0.05 |

more attention to newer features when encountering similar historical memories. In Table G, we experimented with not using positional information and using absolute positional encoding. The results show that ALiBi performs the best, while absolute positional encoding performs the worst. This is because we only sample 5 frames as search frames during training, requiring the introduction of positional encoding with extrapolation capability.

Table G: Ablation studies on position bias added to attention score in MCP.

| # | Variants | LaSOT | LasHeR | VisEvent | DepthTrack | $\Delta$ |
|---|---|---|---|---|---|---|
| 1 | ALiBi | 74.7 | 61.2 | 64.2 | 65.9 | 0 |
| 2 | *w/o* Position | 74.6 | 61.2 | 64.0 | 65.9 | **-0.08** |
| 3 | Absolute Position Encoding | 74.4 | 61.0 | 64.1 | 65.9 | **-0.15** |

**Select Strategy of Memory.** Our current memory bank selection strategy is to select n frames uniformly at equal intervals from all previously tracked frames. In addition, we also experimented with other selection strategies, with the results presented in Table H. We try adding search region features to the memory bank every 5 frames and 10 frames while using a first-in-first-out (FIFO) strategy to maintain the memory bank size. The results show that uniform sampling can ensure longer-term memory and yields significant benefits, especially on long-term tracking datasets such as LaSOT.

# B  B. MORE DETAILED RESULTS IN DIFFERENT ATTRIBUTE SCENES ON LASOT

LaSOT (Fan et al., 2019) is well-known for featuring a diverse range of challenging tracking scenarios, therefore, in Figure A, we provide a more detailed comparison of our proposed Uni-MDTrack-B with other current excellent trackers MambaLCT$_{256}$ (Li et al., 2025), LoRAT-B$_{378}$ (Lin et al., 2025), HIPTrack (Cai et al., 2024), ARTrackV2 (Bai et al., 2024), and OSTrack (Ye et al., 2022) across various challenging scenario subsets in LaSOT (Fan et al., 2019). Figure A presents detailed success curves and AUC scores across individual subsets, along with the success and precision curves on the entire LaSOT *test* split. The results demonstrate that our Uni-MDTrack significantly outperforms these RGB-based trackers both overall and across the vast majority of subsets.

# C  C. MORE QUALITATIVE RESULTS

## C.1  RESULTS ON RGB VISUAL TRACKING

In order to visually highlight the advantages of our method over existing approaches in challenging scenarios, we provide the detail visualization results in Figure B. All videos are from the *test* split of LaSOT. We compare our proposed Uni-MDTrack-B with SUTrack-B(Chen et al., 2025) and MambaLCT$_{256}$ (Li et al., 2025) in terms of performance when the target undergoes sudden movement, deformation, occlusion, and scale variation. All the selected videos are challenging, as described below:

- Figure B(a) demonstrates the tracking results of three methods when the target suffer from large scale variations.

- Figure B(b) demonstrates the tracking results of three methods when the targets have partial occlusions and among similar objects.

Table H: Ablation studies on memory select strategy.

| # | Variants | LaSOT | LasHeR | VisEvent | DepthTrack | Δ |
|---|----------|-------|--------|----------|------------|---|
| **1** | **Ours** | 74.7 | 61.2 | 64.2 | 65.9 | 0 |
| **2** | FIFO + 5 Interval | 74.1 | 61.1 | 63.8 | 65.8 | **-0.3** |
| **3** | FIFO + 10 Interval | 74.2 | 61.1 | 63.9 | 65.9 | **-0.23** |

- Figure B(c) demonstrates the tracking results of three methods when the target suffers sudden movement or occlusion.

We observe that in large scale variations (as shown in Figure B(a)), previous trackers struggle to maintain consistent tracking of the correct target. In contrast, our Uni-MDTrack demonstrates superior performance in accurately identifying and consistently tracking the target, even in the presence of sudden movements (as illustrated in Figure B(c)). Additionally, in Figure B(b), Uni-MDTrack can effectively discriminate the background distractors.

Although our method demonstrates clear advantages over prior approaches, tracking failures can still occur in extremely challenging scenarios. As shown in Figure F, when the coin is heavily occluded and surrounded by numerous visually similar distractors, all methods eventually fail; however, our method is still able to track the target during the initial stage of occlusion. A similar situation arises in the racing scenario, where the background contains many nearly identical cars that frequently overlap, making the target extremely difficult to predict. In the volleyball scenario, the ball undergoes rapid motion, appears small in scale, and is often blurred, which likewise makes prediction highly challenging. These types of scenes are universally difficult for current tracking models.

## C.2 RESULTS ON MULTI-MODAL VISUAL TRACKING

We further provide visualized comparisons of our proposed Uni-MDTrack against other excellent trackers SUTrack (Chen et al., 2025), and STTrack (Hu et al., 2025) across other modalities including RGB-Depth in Figure C(a), RGB-Event in Figure C(b) and RGB-Thermal in Figure C(c). Our Uni-MDTrack consistently exhibit superior performance on these modalities.

## C.3 ATTENTION MAP COMPARISON OF TEMPORAL PROPAGATE TOKEN AND DYNAMIC STATE FEATURE

In Figure D, we visualized the attention maps for the representative method ODTrack (Zheng et al., 2024), which uses Temporal Propagate Tokens, and compared the attention maps with our approach. For ODTrack, since the Temporal Propagate Token, template, and search region tokens are concatenated together, we can visualize the attention of the Temporal Propagate Token to the template and search region to ascertain what information it actually integrates. On the left side of Figure D, we sum and normalize the attention weights of each layer in ODTrack-B, revealing that a significant portion of the Temporal Propagate Token's attention is focused on the templates. This can hinder the token's ability to integrate information about target dynamic state changes. On the right side of Figure D, we visualize the cross-attention map of our search region to the Dynamic State Feature in DSF, showing that the attention is more focused and precise.

## C.4 VISUALIZATION OF THE DYNAMIC STATE FEATURES OUTPUT BY DSF ON THE SEARCH REGION WHEN THE TARGET UNDERGOES SCALE VARIATION AND FAST MOTION

To further illustrate how the dynamic target state encoded by DSF evolves when the target undergoes short-term scale variation or fast motion, we select two representative scenarios in which the target experiences noticeable scale variation and fast movement. As shown in Figure E, we visualize the averaged cross-attention weights between the outputs of all DSF modules and the search-region features. Compared with Figure D, we increase the transparency of the heatmaps and use consecutive frames to make the target's motion or appearance changes easier to observe.

From the results in Figure E, for the eagle with unfolding wings, the dynamic features produced by DSF clearly focus on the motion of the wings as they extend. For the fast-moving airplane, the dynamic features output by DSF clearly focus on the airplane's head-to-tail motion direction. When

the airplane's head is occluded, the attention map can roughly estimate the head position, and when the airplane reveals previously occluded parts, DSF can quickly allocate attention to these emerging regions.

