# OpenReview forum: "Uni-MDTrack: Prompt Unified Single Object Tracking with Deep Fusion of Memory and Dynamic State"
_ICLR.cc/2026/Conference — Submitted to ICLR 2026_

### Official Review · Reviewer_aS26 · 2025-10-31

**Soundness:** 3
**Presentation:** 3
**Contribution:** 3
**Rating:** 6
**Confidence:** 5

**Summary:**

This work presents a parameter-efficient fine-tuning (PEFT) framework for unified single-object tracking across multiple modalities (RGB, RGB-T, RGB-D, RGB-E, RGB-Language). The method introduces two novel components: a Memory-Aware Compression Prompt (MCP) module and Dynamic State Fusion (DSF) modules. MCP maintains a fixed set of learnable query tokens that dynamically compress the historical memory bank into a small set of prompt tokens. These tokens are concatenated to the input sequence, acting as additional prompts that inject long-term contextual memory into the backbone in a parameter-efficient way. DSF uses a selective state-space model (SSM) with gating to capture continuous target dynamics across layers. The experiments and ablations confirm efficiency and generalizability.

**Strengths:**

1. Novel architecture that merges memory compression and state-space modeling for efficient and unified multimodal tracking across diverse modalities.
2. Parameter efficiency: The base version (Uni-MDTrack-B) fine-tunes only 30 % of its total parameters while achieving comparable or better accuracy than other full-tuning baselines.
3. Comprehensive experiments and ablations demonstrating component effectiveness and scalability.

**Weaknesses:**

1. Although the paper reports reduced GFLOPs and parameter counts, it does not provide the actual inference speed (FPS). Given that the backbone is HiViT and additional modules (MCP and DSF) involve memory querying and state updates, the real-time performance might be limited.
2. The DSF module employs an SSM formulation that is similar to the Mamba architecture, including the gating and state update mechanisms. While this integration into a tracking framework is reasonable, the design itself does not introduce substantial architectural novelty.

**Questions:**

1. Could the authors provide the inference time (FPS) and runtime complexity comparisons with existing baselines (e.g., SUTrack, HIPTrack)? This would better substantiate the efficiency claims of Uni-MDTrack.
2. The DSF module seems structurally similar to Mamba’s selective state-space formulation. Could the authors clarify what specific modifications or adaptations were made beyond directly applying the Mamba equations?
3. The paper already covers various benchmarks (LaSOT, LasHeR, VisEvent, DepthTrack). Could the authors further discuss or analyze how Uni-MDTrack behaves in long-term tracking situations (e.g., target reappearance, heavy occlusion) to demonstrate temporal robustness?

---

> ### Author Response · Authors · 2025-11-28
> **Response to Reviewer aS26 (Part 1)**
>
> We sincerely appreciate your constructive feedback, and we are greatly encouraged by your positive assessment of our work. Below are our responses, which we hope will satisfactorily address your concerns.
>
> ---
>
> > W1:  Although the paper reports reduced GFLOPs and parameter counts, it does not provide the actual inference speed (FPS). Given that the backbone is HiViT and additional modules (MCP and DSF) involve memory querying and state updates, the real-time performance might be limited.
> >
>
> `Response:` Thank you for your suggestion. Actually, inference speed is affected by numerous factors, such as IO speed, CPU and GPU performance. Since different methods report speeds tested on different devices, we believe computational metrics provide a fairer comparison, which is why we did not originally report speed. However, we have now conducted speed testing of our method. On A100 GPU and Intel(R) Xeon(R) Platinum 8369B CPU, Uni-MDTrack-B224 achieved **62  FPS**, while Uni-MDTrack-L384 achieved **15 FPS**.
>
> Since our memory size is fixed, we can keep it permanently on the GPU without introducing significant memory‑access overhead. Moreover, our state update does not require the complex scanning procedures used in prior methods that employ SSMs as backbone network layers. Instead, the state only needs to be updated once after processing each frame. Therefore, these operations do not introduce any noticeable slowdown to our method.
>
> ---
>
> > W2:  The DSF module employs an SSM formulation that is similar to the Mamba architecture, including the gating and state update mechanisms. While this integration into a tracking framework is reasonable, the design itself does not introduce substantial architectural novelty.
> >
>
> `Response:` We are more than willing to reiterate the innovation and significance of our method here. As we responded to Reviewer `6Uu5` `W1`, there are significant differences between our DSF module and the usage of previous SSM modules.
>
> Previous methods such as MambaVT [1] and MambaVLT [2] predominantly use SSM as the backbone network implementation, or replace certain layers of the backbone with SSM, fundamentally playing the same role as other backbone layers while merely utilizing SSM's linear computational complexity to extend context length. These approaches also require designing complex scanning algorithms. But linear SSMs are not guaranteed to outperform Transformers under the same sequence length (as proven in [3]). We have conducted experiments demonstrating this limitation, where replacing our DSF module with the same number of encoder layers from MambaVT and inserting them at regular intervals throughout the backbone network, which results in substantial performance degradation, as shown in below Table.
>
> | Model | SSM in DSF Module (ours) | SSM as Backbone Layer |
> | --- | --- | --- |
> | LaSOT AUC | 74.7 | 74.0 |
> | VisEvent | 64.2 | 63.5 |
> | LasHeR | 61.2 | 60.5 |
> | DepthTrack | 65.9 | 65.4 |
>
> In contrast, our DSF module serves a fundamentally different purpose. Unlike previous SSM methods that require designing complex scanning sequences and methodologies, DSF continuously utilizes new frame search region features to update target states and employs these states as supplementary information for the backbone network. The key focus of the DSF module is to demonstrate that **continuously updated dynamic target states can serve as efficient and effective complements to the backbone network, rather than being constrained to specific model designs**. Additionally, results from Table D in our appendix indicate that although replacing SSM with other recurrent structures (e.g., LSTM) causes a certain degree of performance loss, there is still a gain compared to not using such structures.
>
> Furthermore, rather than re‑architecting or replacing backbone layers, we inject the dynamic states with DSF in **a lightweight adapter manner into a frozen HiViT backbone**. As Reviewer `gefL` noted, the overall design is “technically neat” and “uncommon.” The same adapters plug into other trackers and yield consistent gains across 11 benchmarks on SUTrack (Table 1, Uni‑MDTrack), SPMTrack (Table 2, MDTrack), and DropTrack (Table 7, DropTrack w/ Ours) with minimal parameters that need to be retrained and no change to their core architectures.
>
> [1] Lai S, Liu C, Zhu J, et al. Mambavt: Spatio-temporal contextual modeling for robust rgb-t tracking[J]. IEEE Transactions on Circuits and Systems for Video Technology, 2025.
>
> [2] Liu X, Zhou L, Zhou Z, et al. Mambavlt: Time-evolving multimodal state space model for vision-language tracking[C]//Proceedings of the Computer Vision and Pattern Recognition Conference. 2025: 8731-8741.
>
> [3] Merrill W, Petty J, Sabharwal A. The illusion of state in state-space models[C]//Proceedings of the 41st International Conference on Machine Learning. 2024: 35492-35506.

---

> ### Author Response · Authors · 2025-11-28
> **Response to Reviewer aS26 (Part 2)**
>
> ---
>
> > Q1:  Could the authors provide the inference time (FPS) and runtime complexity comparisons with existing baselines (e.g., SUTrack, HIPTrack)? This would better substantiate the efficiency claims of Uni-MDTrack.
> >
>
> `Response:` We have already reported the runtime speed of our method in our response to `W1`. In addition, Table 1 in the main paper provides a detailed comparison of the computational cost and the number of parameters of our method.
>
> We would like to respectfully express our perspective that, computational cost is a relatively fair indicator for measuring efficiency in visual tracking, because under the same hardware configuration, models with similar computational requirements that exhibit significant speed differences are generally limited by memory access bottlenecks. Trackers, unless they maintain an extremely large context, typically do not involve time-consuming memory access operations. For memory access operations within the GPU, optimization tools such as Triton can fuse operations into a single kernel function, thus making computational cost a more critical indicator.
>
> ---
>
> > Q2:  The DSF module seems structurally similar to Mamba’s selective state-space formulation. Could the authors clarify what specific modifications or adaptations were made beyond directly applying the Mamba equations?
> >
>
> `Response:` We have answered this question in our response to `W2`.
>
> ---
>
> > Q3:  The paper already covers various benchmarks (LaSOT, LasHeR, VisEvent, DepthTrack). Could the authors further discuss or analyze how Uni-MDTrack behaves in long-term tracking situations (e.g., target reappearance, heavy occlusion) to demonstrate temporal robustness?
> >
>
> `Response:` We fully understand your concern regarding the long-term tracking capability. In fact, LaSOT and LaSOT_ext are themselves large-scale long-term tracking datasets—each video typically contains over 2000 frames, with some exceeding 4000 frames.  In comparison, GOT-10K generally contains only tens of frames, while TrackingNet typically has at most hundreds of frames. As shown in Table 3, our method outperforms SUTrack-B224 on LaSOT by +1.5 AUC and outperforms SUTrack-L384 by +0.9 AUC. Furthermore, DropTrack equipped with our approach (Table 7) achieves a +1.3 AUC gain on LaSOT. Our method also demonstrates very significant gains on LaSOT_ext, with these results clearly demonstrating the effectiveness of our approach in long-term tracking tasks.
>
> Meanwhile, in Figure A of Appendix B, we provide a detailed visualization of the performance curves of various methods across each special scenario subset of the LaSOT test set. Specifically, our method significantly outperforms other methods in variation scenarios such as Deformation and Scale Variation, as well as in occlusion scenarios including Partial Occlusion and Full Occlusion.
>
> Figure B(c) highlights two challenging long-term scenarios under heavy occlusion. In the first row, a zebra is fully occluded by a large horse at frame #0248, reappears at frame #0324, and appears again at frame #1257. MambaLCT drifts immediately after the first reappearance, and SUTrack fails at frame #1257, whereas our method remains consistently aligned with the ground truth throughout. In the second row, our model reliably tracks the target ball from frame #0244 to #1378, while both SUTrack and MambaLCT lose the target early due to occlusion at frame #0351. These results collectively demonstrate that our approach significantly enhances long-term stability in challenging occlusion and re-appearance scenarios.

---

### Official Review · Reviewer_sPSa · 2025-10-31

**Soundness:** 3
**Presentation:** 2
**Contribution:** 2
**Rating:** 4
**Confidence:** 5

**Summary:**

This paper introduces Uni-MDTrack, a parameter-efficient fine-tuning (PEFT) framework for single object tracking, focusing on the trending unified trackers. The core of the method lies in two novel, lightweight modules designed to enhance a frozen foundation model:
1. Memory-Aware Compression Prompt (MCP): Efficiently compresses a long-term memory bank into a few prompt tokens using trainable queries. These tokens are prepended to the input sequence, enabling deep interaction with image features throughout the backbone.
2. Dynamic State Fusion (DSF): Uses a State-Space Model (SSM) to capture the target's continuous motion dynamics. The resulting state features are fused at multiple levels of the backbone.

By fine-tuning only these modules and the prediction head (~30% of parameters), the model achieves state-of-the-art performance across 10 multi-modal tracking datasets.

**Strengths:**

-Design: MCP offers a good way to integrate long-term memory without the high cost of long sequences. DSF's use of an SSM to model dynamics is modern and its multi-level fusion is effective.
-Performance: The results are very good. The method sets a new SOTA on a wide range of challenging benchmarks (LaSOT, TrackingNet, TNL2K, etc.), with significant gains over a very strong baseline (SUTrack).
-Evaluation: The ablation studies are comprehensive and convincingly validate the design choices behind both modules. The authors clearly demonstrate the individual contributions of MCP and DSF and justify their architectural decisions.
-Generalizability: The modules are shown to be plug-and-play, boosting the performance of another tracker (DropTrack) and outperforming competing PEFT methods. This highlights their general utility.

**Weaknesses:**

-Limited Mathematical Analysis: The level of mathematical grounding of the ideas in the paper is somewhat below ICLR standards. Although the equations provided description of the proposed modules, the ICLR standards assume theoretical mathematical analysis of “why” these modules work, in addition to the empirical evidence. See previous closely related tracking papers in ICLR2024 and 2024.
-No Inference Speed (FPS): The paper reports FLOPS but omits FPS, which is a critical metric for any tracker. Without it, the practical performance-efficiency trade-off is unclear.
-Limited Failure Analysis: While successes are shown, a discussion of the method's failure cases would provide more complete insight into its limitations.
- Relies on a Strong Baseline: The method is built on the powerful SUTrack model. While the gains are clear, the remarkable performance is a combination of the strong baseline and the proposed modules. A practical reform for this problem is to plug into other trackers (e.g., OSTrack, ODTrack, …) that have a more vanilla flavour of pure transformers.

**Questions:**

see above in weeknesses

---

> ### Author Response · Authors · 2025-12-01
> **Response to Reviewer sPSa (Part 1)**
>
> We sincerely thank the reviewer for the feedback. Below are our responses, which we hope will satisfactorily address your concerns.
>
> ---
>
> > W1:  Limited Mathematical Analysis: The level of mathematical grounding of the ideas in the paper is somewhat below ICLR standards. Although the equations provided description of the proposed modules, the ICLR standards assume theoretical mathematical analysis of “why” these modules work, in addition to the empirical evidence. See previous closely related tracking papers in ICLR2024 and 2024.
> >
>
> `Response:`
>
> Thank you for your suggestion on further strengthening our paper. We would like to respectfully express a point of confusion. To the best of our knowledge, ICLR does not explicitly require mathematical derivations, and our submission falls under the track of “applications to computer vision, audio, language, and other modalities,” rather than machine learning theory.
>
> Although we cannot engage in multi-round discussions at this time, we would also be grateful if you could kindly provide the specific references for the *“ICLR Standards”* and *“previous closely related tracking papers in ICLR 2024 and 2024”* that you mentioned in final comment. This would be extremely helpful for us.
>
> That said, we remain very willing to conduct further analysis of our method, as described below:
>
> - **Analysis 1: The Limitations of Pure SSMs in Long-Sequence Extrapolation**
>
>     Our method uses SSM to model the continuous dynamic features of targets, which differs from previous methods that use SSM as the backbone network by leveraging processing efficiency to extend sequence length, as we responded to Reviewer `6Uu5` `W1`.
>
>     Recent work also challenges the prevailing belief that Mamba-style State-Space Models "fundamentally solve" long-sequence modeling. [1] demonstrate that standard deep SSMs and Transformers share similar state-tracking limitations (both residing in $\mathsf{TC}^0$), implying that simply maintaining a recurrent "state" is insufficient to overcome the core difficulties of long-range tracking. Furthermore, empirical analyses such as *LongMamba* [2] and *ReMamba* [3] reveal a critical flaw: even after long-context fine-tuning, Mamba struggles to extrapolate effectively, often lagging behind Transformers as sequence length grows.
>     Formally, the hidden state update in Mamba is defined as:
>     $H_t = \bar A_t \odot H_{t-1} + \bar B_t \odot X_t,$ where  $\bar A_t$ represents the channel-wise decay factors, and each element in  $\bar A_t \in (0,1)$. A key structural constraint lies in the parameterization of $\bar A_t$:
>     $\Delta_t = \mathrm{Softplus}(X_t), \qquad \bar A_t = \exp(\Delta_t \odot A),$
>     with $A<0$ being a fixed learnable matrix. Consequently, every dimension of $\bar A_t$ is strictly less than 1. By unrolling the recurrence, the contribution of an early token $X_j$ to the output at position $i$ is proportional to the cumulative product of decay factors:
>     $\alpha_{i,j} \;\propto\; \left(\prod_{k=j+1}^{i}\bar A_k\right)\odot \bar B_j.$
>     Leveraging $\bar A_k = \exp(\Delta_k\odot A)$, this product collapses into a unified exponential term:
>     $\prod_{k=j+1}^{i}\bar A_k = \exp\left[ \left(\sum_{k=j+1}^{i}\Delta_k\right)\odot A \right]$.
>     Since $A<0$ and $\Delta_k>0$, for any "global channel" dimension, there exists a constant $c>0$ such that the magnitude of influence decays exponentially with distance:
>     $\left\|\prod_{k=j+1}^{i}\bar A_k\right\| \le \exp\big(-c (i-j)\big).$
>     This derivation leads to a direct conclusion: the influence of early tokens vanishes exponentially as the distance $i-j$ increases. While this decay may be manageable within the training length $L_{\text{train}}$, it becomes catastrophic during inference when extrapolating to $L_{\text{test}} \gg L_{\text{train}}$.  Therefore, directly using an SSM as the backbone and simply extending the context length—as done in prior methods—will inevitably diminish the influence of earlier tokens. This is precisely why we employ an SSM‑like structure only within DSF to model dynamic features, rather than introducing long‑term memory.
>
> [1] Merrill W, Petty J, Sabharwal A. The illusion of state in state-space models[C]//Proceedings of the 41st International Conference on Machine Learning. 2024: 35492-35506.
>
> [2] Ye Z, Xia K, Fu Y, et al. LongMamba: Enhancing Mamba's Long-Context Capabilities via Training-Free Receptive Field Enlargement[C]//The Thirteenth International Conference on Learning Representations.
>
> [3] Yuan D, Liu J, Li B, et al. Remamba: Equip mamba with effective long-sequence modeling[J]. arXiv preprint arXiv:2408.15496, 2024.

---

> ### Author Response · Authors · 2025-12-01
> **Response to Reviewer sPSa (Part 2)**
>
> ---
>
> > W1:  Limited Mathematical Analysis.
> >
>
> `Response:`
> - **Analysis 2: Our Approach: Hybrid Dynamic States with Memory**
>
>     In contrast to previous methods that leverage SSMs, we deliberately avoid relying on a single compressed state to store all historical information. Our architecture shifts the burden of hard state tracking to a hybrid design combining **DSF** and **MCP**.
>
>     From our previous derivation, the influence of early tokens vanishes exponentially as the distance increases. Therefore, we do not ask a single backbone to satisfy handling very long spatio-temporal contexts. Instead, we decouple the task based on temporal distance:
>
>     1.  Short-range dynamic state modeling: Handled by the DSF module. This acts as a lightweight, looped recurrent adapter operating on a frozen HiViT backbone. It utilizes the SSM's exponential decay dynamics, which aligns perfectly with the natural locality of immediate motion continuity (e.g., physics-based object inertia), and plays a role of a “reasoner”. As shown in Figure E in our revised appendix, the dynamic features produced by DSF tend to focus on the motion tendencies of the target and on regions where noticeable changes occur, such as the unfolding wings of the eagle and the nose and tail of the fast‑moving airplane.
>     2.  Long-range dependencies: Handled by the MCP module. This module overcomes the bottleneck of the SSM-like MCP structure via direct retrieval, maintaining an explicit memory bank to handle global associations (e.g., re-identifying the target after long-term occlusion).
>
>     This decomposition allows DSF to provide stable, continuous dynamics for local consistency, while MCP’s distance-aware attention (equipped with ALiBi) robustly handles global context. This explains our empirical superiority in extending to long video sequences compared to simply scaling up Mamba-style models.
>
>     **DSF for Local Dynamics:**  Similar to Mamba, our DSF block utilizes exponential decay dynamics. However, we restrict its role strictly to modeling local state transitions (e.g., immediate motion continuity in video tracking). By limiting the scope, the exponential decay aligns with the natural locality of short-term motion, avoiding the loss of long-term semantic information. Unlike standard paradigms that burden the visual backbone with temporal fusion, we design the DSF as a decoupled, lightweight recurrent adapter. It operates on a frozen HiViT backbone, inserting essential temporal-reasoning capacity through a looped connection without retraining the heavy visual encoder.
>
>     **MCP for Long-Range Associations:** To handle long-range dependencies—such as re-identifying a template after occlusion—we employ the MCP module. This module maintains an explicit memory bank of past search features, compressed into a fixed number of tokens via attention with ALiBi biases.
>     Let the current query be $q_t$ and the memory bank contain keys $(k_i)$ indexed by $i \in \mathcal{M}$. The attention scores are defined as:
>     $a_{t,i} = \frac{q_t^\top k_i}{\sqrt d}+\beta\,\Delta(i,t), \qquad \beta < 0,$
>     where $\Delta(i,t)$ is the temporal distance, and $\beta$ is ALibi slope.
>     Thus, the cross-attention logit and softmax weight are
>     $p_{t,i} = \frac{e^{a_{t,i}}}{\sum_{j\in\mathcal M} e^{a_{t,j}}}.$
>
>     **1. Explicit Recency Bias:**
>     Considering a simplified case where feature similarities are negligible ($q_t^\top k_i \approx q_t^\top k_j=0$), the relative attention weight between two memories $i$ and $j$ depends solely on their distance:
>     $\frac{p_{t,i}}{p_{t,j}} = \exp\big(\beta[\Delta(i,t)-\Delta(j,t)]\big).$
>     If memory $i$ is more recent than $j$ (i.e., $\Delta(i,t) < \Delta(j,t)$), then $p_{t,i} > p_{t,j}$. This provides a clean, explainable mechanism for prioritizing recent observations.
>
>     **2. Bounded Extrapolation Error:**
>     Assume the model is trained with a memory length $K$, but tested with length $L > K$. The total attention mass contributed by the unseen "tail" (memories beyond distance $K$) is bounded by a geometric series:
>     $\text{Mass}_{\text{tail}} = \sum _{k=K+1}^{L} e^{\beta k} <  \sum _{k=K+1}^{\infty} e^{\beta k} = \frac{e^{\beta(K+1)}}{1-e^{\beta}}$. To ensure the trained model's attention distribution remains valid during inference, we require this tail mass to be negligible (less than a threshold $\eta$). Solving $\frac{e^{\beta(K+1)}}{1-e^{\beta}} \le \eta$ for $K$ yields:
>     $K \gtrsim \frac{\ln(1/\eta)}{|\beta|} - 1.$
>     This result implies that the effective memory horizon is of order $O(1/|\beta|)$. Consequently, extending the memory bank at test time adds only an exponentially small tail to the distribution, ensuring that our MCP module extrapolates robustly without the catastrophic failure modes observed in pure SSMs.

---

> ### Author Response · Authors · 2025-12-01
> **Response to Reviewer sPSa (Part 3)**
>
> ---
>
> > W2: No Inference Speed (FPS): The paper reports FLOPS but omits FPS, which is a critical metric for any tracker. Without it, the practical performance-efficiency trade-off is unclear.
> >
>
> `Response:` Thank you for your suggestion. Actually, inference speed is affected by numerous factors, such as IO speed, CPU and GPU performance. Since different methods report speeds tested on different devices, we believe computational metrics provide a fairer comparison, which is why we did not originally report speed. However, we have now conducted speed testing of our method. On A100 GPU and Intel(R) Xeon(R) Platinum 8369B CPU, Uni-MDTrack-B224 achieved **62  FPS**, while Uni-MDTrack-L384 achieved **15 FPS**.
>
> > W3: Limited Failure Analysis: While successes are shown, a discussion of the method's failure cases would provide more complete insight into its limitations.
> >
>
> `Response:` Thank you for your suggestion. In the revised appendix, we include several extremely challenging tracking scenarios in Figure F, where all trackers exhibit failure cases under such complex conditions. Although our method demonstrates clear advantages over prior approaches, tracking failures can still occur in these extremely challenging scenarios. As shown in Figure F, when the coin is heavily occluded and surrounded by numerous visually similar distractors, all methods eventually fail; however, our method is still able to track the target during the initial stage of occlusion. A similar situation arises in the racing scenario, where the background contains many nearly identical cars that frequently overlap, making the target extremely difficult to predict. In the volleyball scenario, the ball undergoes rapid motion, appears small in scale, and is often blurred, which likewise makes prediction highly challenging. These types of scenes are universally difficult for current tracking models.
>
> ---
>
> > W4: Relies on a Strong Baseline: The method is built on the powerful SUTrack model. While the gains are clear, the remarkable performance is a combination of the strong baseline and the proposed modules. A practical reform for this problem is to plug into other trackers (e.g., OSTrack, ODTrack, …) that have a more vanilla flavour of pure transformers.
> >
>
> `Response:` We respectfully hope to clarify that the effectiveness of our proposed method is not directly dependent on the baseline choice. Although we adopt SUTrack as our baseline, we are still able to achieve very substantial performance improvements on this foundation. As demonstrated in Tables 2-6, our method achieves improvements exceeding +1.x AUC compared to SUTrack on LaSOT, TrackingNet, LaSOT ext, VisEvent, and LasHeR, and even substantially surpasses SUTrack on TNL2K with a +2.5 AUC improvement, which represents very significant gains in visual tracking tasks. Even the individual application of MCP or DSF modules yielding +0.6 to +0.7 AUC improvements is already highly significant, as shown in Table 6. We believe that achieving breakthroughs at a higher level of performance is more meaningful than obtaining improvements on poorer baselines.
>
> On the other hand, we have already validated our method on multiple baselines beyond SUTrack. MDTrack-B on SPMTrack-B (Table 3) yields +0.7 AUC on LaSOT, and DropTrack w/ Ours (Table 7) achieves +1.3 AUC on LaSOT. DropTrack has the same model structure with OSTrack. These results show that our modules deliver consistent, plug-and-play improvements across diverse and more vanilla transformer-based baselines.

---

### Official Review · Reviewer_gefL · 2025-11-01

**Soundness:** 3
**Presentation:** 3
**Contribution:** 3
**Rating:** 6
**Confidence:** 4

**Summary:**

The paper proposes Uni-MDTrack, a parameter-efficient way to upgrade an existing unified multimodal tracker (like SUTrack) without full retraining. It adds two modules: MCP, which turns a long history of frames into a small set of memory tokens and feeds them through all backbone layers, and DSF, which uses an SSM to update the target’s state from current search features and inject it at multiple depths.

**Strengths:**

Overall, the method is well aligned with current unified-tracking practice and gives a concrete, efficiency-oriented recipe.

- i) The problem setting is realistic: starting from an already unified tracke, how to add history and dynamics without full retraining is a reasonable answer to that question.

ii)  Unlike temporal-propagation tokens that tend to look back at the template, DSF updates state only from search features and injects it at 4 depth levels, which is a good design for capturing short-term pose changes the memory can’t see. Using an SSM here is technically neat and, as the authors claim, uncommon in PEFT tracking.

**Weaknesses:**

- i) MCP uses ALiBi and caps memory at 50 frames by uniform sampling, but we don’t see an analysis of how sensitive performance is to memory length or to distractors

- ii) The paper claims DSF captures “continuous target variation,” but there is no figure/table showing SSM hidden-state evolution vs. scale change / fast motion, so right now we have to trust the ablation. A small visualization would make the SSM choice more convincing.

**Questions:**

i) Can you specify the per-modality sampling ratios during the 100k-sequence epochs? E.g., what percentage is RGB-only vs. RGB-T vs. RGB-D vs. RGB-E vs. RGB-Language?

ii) DSF updates only from search tokens to avoid template distraction — did you try an alternative that also conditions on the current template crop? If so, did the SSM start to “forget” dynamics, as you suggest earlier in the paper?

---

> ### Author Response · Authors · 2025-12-01
> **Response to Reviewer gefL**
>
> We sincerely appreciate your reading and constructive feedback, and we are encouraged by your positive assessment of our work. Below are our responses, which we hope will satisfactorily address your concerns.
>
> ---
>
> > W1:  MCP uses ALiBi and caps memory at 50 frames by uniform sampling, but we don’t see an analysis of how sensitive performance is to memory length or to distractors.
> >
>
> `Response:` **Analysis of memory length sensitivity**： We actually provide a detailed analysis of memory-length sensitivity in Table E (Appendix A.3), where we vary the memory bank size (*n* = 10/20/50/100). Performance consistently improves from 10 to 50, and then saturates and slightly fluctuates at 100, e.g., LaSOT 74.3→74.6→74.7→74.6, which supports our choice of a 50-frame memory.
>
> **Analysis of Distractors**：Thank you for your suggestion! We respectfully hope to clarify that in the evaluation of LaSOT different attribute subsets in Appendix Figure A, we have already included numerous subsets containing distractor scenarios, specifically the '**Background Clutter**' subset, which contains examples such as 'Bird-x' with multiple similar bird animals mutually interfering and occluding each other, and 'coin-x' with many highly similar coins interfering with each other. The results in Figure A demonstrate that our method significantly outperforms previous methods in distractor scenarios.
>
> ---
>
> > W2:  The paper claims DSF captures “continuous target variation,” but there is no figure/table showing SSM hidden-state evolution vs. scale change / fast motion, so right now we have to trust the ablation. A small visualization would make the SSM choice more convincing.
> >
>
> `Response:` Thank you very much for your suggestion! We fully agree with your viewpoint. Since visualizing hidden states alone cannot effectively demonstrate their connection to the search region and lacks intuitiveness, we still choose to visualize the mean cross-attention weights between all DSF module outputs and search region features under scale change and fast motion conditions. To facilitate observing target motion or changes, we increased the heatmap transparency and selected consecutive frames for visualization, as shown in Figure E in our revised appendix.
>
> From the results in Figure E, for an eagle spreading its wings, the dynamic features output by DSF clearly focus on the dynamically opening wings; while for a fast-moving airplane, the dynamic features output by DSF clearly focus on the airplane's head-to-tail motion direction. When the airplane's head is occluded, the attention map can roughly estimate the head position, and when the airplane reveals previously occluded parts, DSF can quickly allocate attention to these emerging regions.
>
> ---
>
> > Q1:  Can you specify the per-modality sampling ratios during the 100k-sequence epochs? E.g., what percentage is RGB-only vs. RGB-T vs. RGB-D vs. RGB-E vs. RGB-Language?
> >
>
> `Response:` Thank you for your question. Regarding the sampling ratios for different modalities, we follow exactly the same strategy as the baseline SUTrack. Specifically, the training datasets we use include LaSOT, GOT‑10K, COCO, TrackingNet, VastTrack, TNL2K, DepthTrack, VisEvent, and LasHeR. During the sampling of each sample, we randomly select the corresponding dataset following the probability ratio of 2:2:2:2:2:2:1:1:1. Converted into modality‑level proportions, the sampling ratio corresponds approximately to RGB : RGB‑L : RGB‑D : RGB‑E : RGB‑T = 10 : 2 : 1 : 1 : 1.
>
> ---
>
> > Q2:  DSF updates only from search tokens to avoid template distraction — did you try an alternative that also conditions on the current template crop? If so, did the SSM start to “forget” dynamics, as you suggest earlier in the paper?
> >
>
> `Response:` Yes, we try this alternative and provide ablation result in Table C. The variant “Output Whole Sequence”, which conditions the SSM update on both the template crop and the search tokens, performs consistently worse than “Output Search Feature,” where the SSM is updated only from search tokens. This supports our claim that incorporating template features introduces template-driven distractions and degrades the SSM’s ability to model continuous target dynamic states.
>
> Additional evidence is provided in Figure D. Our method keeps attention tightly focused within the search region, whereas ODTrack (which processing the entire sequence) frequently attends to irrelevant or other similar objects. This further illustrates how restricting SSM updates to search tokens helps maintain robust and distraction-resistant dynamic state modeling.

---

### Official Review · Reviewer_6Uu5 · 2025-11-01

**Soundness:** 3
**Presentation:** 3
**Contribution:** 2
**Rating:** 4
**Confidence:** 4

**Summary:**

This paper proposes a parameter-efficient fine-tuning framework for unified single-object tracking built around two modules: Memory-Aware Compression Prompt (MCP), which compresses memory features into prompt tokens that interact with the backbone, and Dynamic State Fusion (DSF), which progressively injects continuous target dynamics from shallow to deep layers. Combined into Uni-MDTrack, the approach maintains low computational cost, trains only ~30% of parameters, supports five modalities, and achieves state-of-the-art results across multiple datasets, while the MCP and DSF modules are plug-and-play and broadly applicable.

**Strengths:**

- The paper is clearly written and very easy to understand, with a detailed experimental setup.
- The experimental results are strong, outperforming existing methods on multiple datasets.

**Weaknesses:**

- I sense very limited novelty and contribution to the field, and may not meet the bar for ICLR. The partial fusion of SSM with a Transformer backbone and the use of memory tokens have already been explored in many prior works. Moreover, Table 6's ablation results suggest MCP and DSF do not provide substantial gains—those improvements could likely be achieved by hyperparameter tuning. It's also unclear whether replacing DSF with a simpler module would yield similar effects without introducing an SSM.
- The tracking community largely overlooks the impact of large multi-modal models and still compares methods within a small circle. This hinders progress, and the paper should at least compare with or pay more attention to large-model-based tracking approaches.

**Questions:**

see weaknesses

---

> ### Author Response · Authors · 2025-11-28
> **Response to Reviewer 6Uu5 (Part 1)**
>
> We sincerely thank the reviewer for the feedback. Below are our responses, which we hope will satisfactorily address your concerns.
>
> ---
>
> > W1: Limited novelty and contribution to the field. The partial fusion of SSM with a Transformer backbone and the use of memory tokens have already been explored in many prior works.
> >
>
> `Response:` We greatly appreciate your reading of our manuscript and the comments you have provided! We are also more than willing to reiterate the innovation and significance of our method here.
>
> - Compared to previous memory methods, our memory module MCP strikes a better balance among memory richness, interaction sufficiency, and computational efficiency.
>     - Previous methods such as HIPTrack [1], to ensure computational efficiency, only fuse memory features with search region features after the backbone network, and additionally design a dedicated memory feature encoder. This results in more complex network architectures while enabling insufficient fusion between memory features and search region features.
>     - Another prior method, ODTrack [2], concatenates multiple historical frames into the input token sequence. This ensures network simplicity and allows sufficient fusion between historical features and current search region features. However, due to the quadratic complexity of Transformer encoders, only a small number of historical frames can be concatenated, which limits memory richness.
>
>     In contrast, our MCP module compresses rich memory pool information into a fixed number of sparse memory tokens through elegant memory queries. This approach controls input sequence length and maintains computational efficiency while preserving rich memory information. Additionally, it enables deep fusion between memory tokens and search region features within the backbone network. As shown in Table 6, the MCP module achieves an average improvement of +0.6 AUC, with a +0.9 AUC improvement specifically on the long-term tracking dataset LaSOT. In comparison, HIPTrack only achieves +0.9 AUC improvement on LaSOT over its baseline while introducing more parameters and computational overhead.
>
> - There are also significant differences between our DSF module and the usage of previous SSM modules
>     - Previous methods such as MambaVT [3] and MambaVLT [4] predominantly use SSM as the backbone network implementation, or replace certain layers of the backbone with SSM, fundamentally playing the same role as other backbone layers while merely utilizing SSM's linear computational complexity to extend context length. These approaches also require designing complex scanning algorithms. But linear SSMs are not guaranteed to outperform Transformers under the same sequence length (as proven in [5]). We have conducted experiments demonstrating this limitation, where replacing our DSF module with the same number of encoder layers from MambaVT and inserting them at regular intervals throughout the backbone network, which results in substantial performance degradation, as shown in below Table.
>
>
>         | Model | SSM in DSF Module (ours) | SSM as Backbone Layer |
>         | --- | --- | --- |
>         | LaSOT AUC | 74.7 | 74.0 |
>         | VisEvent | 64.2 | 63.5 |
>         | LasHeR | 61.2 | 60.5 |
>         | DepthTrack | 65.9 | 65.4 |
>
>     In contrast, our DSF module serves a fundamentally different purpose. Unlike previous SSM methods that require designing complex scanning sequences and methodologies, DSF continuously utilizes new frame search region features to update target states and employs these states as supplementary information for the backbone network. The key focus of the DSF module is to demonstrate that **continuously updated dynamic target states can serve as efficient and effective complements to the backbone network, rather than being constrained to specific model designs**. Additionally, results from Table D in our appendix indicate that although replacing SSM with other recurrent structures (e.g., LSTM) causes a certain degree of performance loss, there is still a gain compared to not using such structures.
>
> - Furthermore, rather than re‑architecting or replacing backbone layers, we inject memory and dynamic states with MCP and DSF in **a lightweight adapter manner into a frozen HiViT backbone**. As Reviewer `gefL` noted, the overall design is “technically neat” and “uncommon.” The same adapters plug into other trackers and yield consistent gains across 11 benchmarks on SUTrack (Table 1, Uni‑MDTrack), SPMTrack (Table 2, MDTrack), and DropTrack (Table 7, DropTrack w/ Ours) with minimal parameters that need to be retrained and no change to their core architectures.
>
> Additional analysis can also be found in our response to Reviewer `sPSa` `W1`.

---

> ### Author Response · Authors · 2025-11-28
> **Response to Reviewer 6Uu5 (Part 2)**
>
> ---
>
> > W1: Table 6's ablation results suggest MCP and DSF do not provide substantial gains—those improvements could likely be achieved by hyperparameter tuning.
> >
>
> `Response:` We respectfully wish to clarify that the performance gains achieved by our method are indeed substantial. Compared to our baseline SUTrack, we have obtained improvements exceeding +1.x AUC across LaSOT, TrackingNet, LaSOT ext, VisEvent, and LasHeR, and even substantially surpassed +2.5 AUC on TNL2K, which represents very significant gains in visual tracking tasks. Even the individual application of MCP or DSF modules yielding +0.6 to +0.7 AUC improvements is already highly significant. For instance, as shown in Table 3, ARPTrack (2025) achieves almost identical performance to ARTrack (2023); under the same 384 resolution, MambaLCT from 2025 shows only a +0.4 AUC improvement over ODTrack-B from 2024 on LaSOT.  You can judge the significance of our method's gains by examining the comparisons between other methods in Tables 2, 3, 4, and 5.
>
> ---
>
> > W1: It's also unclear whether replacing DSF with a simpler module would yield similar effects without introducing an SSM.
> >
>
> `Response:` As we responded in `W1`, we have already experimented with replacing SSM with other recurrent architectures such as LSTM in Table D of our appendix. While this approach yields smaller benefits compared to SSM due to the inherent limitations of the model architecture itself, the improvements are still measurable.
>
> ---
>
> > W2: The tracking community largely overlooks the impact of large multi-modal models and still compares methods within a small circle. This hinders progress, and the paper should at least compare with or pay more attention to large-model-based tracking approaches.
> >
>
> `Response:` We thank the reviewer for raising this important point. We would like note that the proposed framework already incorporates multi-modal components rather than being a purely single-modal tracker. The multimodal benchmarks we evaluate on, such as LasHeR (RGB-T), VisEvent (RGB-E), DepthTrack (RGB-D), TNL2K (RGB-Language), are all multi-modal tracking datasets and our method obtain superior performance as shown in Table 2-5. At the same time, we also developed Uni‑MDTrack‑L based on the HiViT‑Large backbone, which already represents a large‑scale model in the field of single‑object tracking. Prior methods such as ViPT [6], SDSTrack [7], and OneTracker [8] conduct experiments only on models of the ViT‑B scale.
>
> We guess the reviewer may be referring to incorporating LLM/VLM-style models into tracking. However, the primary goal of visual tracking is to achieve real-time tracking in continuous video sequences as far as possible. Deploying VLM models for real-time tracking is currently impractical. The goal of this paper is to design a lightweight, plug-and-play module that improves tracking performance while preserving efficiency and deployability.
>
> [1] Cai W, Liu Q, Wang Y. Hiptrack: Visual tracking with historical prompts[C]//Proceedings of the IEEE/CVF Conference on Computer Vision and Pattern Recognition. 2024: 19258-19267.
>
> [2] Zheng Y, Zhong B, Liang Q, et al. Odtrack: Online dense temporal token learning for visual tracking[C]//Proceedings of the AAAI conference on artificial intelligence. 2024, 38(7): 7588-7596.
>
> [3] Lai S, Liu C, Zhu J, et al. Mambavt: Spatio-temporal contextual modeling for robust rgb-t tracking[J]. IEEE Transactions on Circuits and Systems for Video Technology, 2025.
>
> [4] Liu X, Zhou L, Zhou Z, et al. Mambavlt: Time-evolving multimodal state space model for vision-language tracking[C]//Proceedings of the Computer Vision and Pattern Recognition Conference. 2025: 8731-8741.
>
> [5] Merrill W, Petty J, Sabharwal A. The illusion of state in state-space models[C]//Proceedings of the 41st International Conference on Machine Learning. 2024: 35492-35506.
>
> [6] Zhu J, Lai S, Chen X, et al. Visual prompt multi-modal tracking[C]//Proceedings of the IEEE/CVF conference on computer vision and pattern recognition. 2023: 9516-9526.
>
> [7] Hou X, Xing J, Qian Y, et al. Sdstrack: Self-distillation symmetric adapter learning for multi-modal visual object tracking[C]//Proceedings of the IEEE/CVF Conference on Computer Vision and Pattern Recognition. 2024: 26551-26561.
>
> [8] Hong L, Yan S, Zhang R, et al. Onetracker: Unifying visual object tracking with foundation models and efficient tuning[C]//Proceedings of the IEEE/CVF conference on computer vision and pattern recognition. 2024: 19079-19091.

---

### Author Response · Authors · 2025-12-03
**Summary of Author Response**

**Dear Area Chairs, Senior Area Chairs, Program Chairs, Reviewers:**

---

We fully understand that the recent technical issues with OpenReview have hindered the interactive discussion that normally takes place during the rebuttal phase. We would like to express our sincere gratitude to the Area Chairs and Program Chairs for their extraordinary efforts under these challenging circumstances, and to the reviewers for their time and for carefully reading our responses. Given that no active discussion has occurred before the incident, we have summarized the key points from the original reviews alongside our responses below. We hope this overview will assist the Area Chairs and alleviate their workload during the final evaluation, and also enable the Reviewers to quickly grasp the essence of our overall rebuttal.

We sincerely thank all reviewers for their positive assessment and recognition of our work. We are encouraged that all reviewers acknowledge the **excellent performance** of our method **across multiple datasets** and agree on the **soundness of our motivation and methodology**. Specifically, Reviewer aS26 comment that our architecture is **novel**, while Reviewer gefL comment that our method is **reasonable and technically neat**. Reviewer 6Uu5 comment that the **presentation of our paper is very good**; and both Reviewer sPSa and Reviewer aS26 remarked that **our experiments are comprehensive and convincing**.

In addition, we have provided **detailed point-by-point responses** to every weakness and question raised by each reviewer in our individual replies, and updated our manuscript (the revised parts are highlighted in cyan). Our rebuttal can be summarized as follows:

- **For Reviewer 6Uu5:**
    - We thoroughly clarify the **significant distinctions** between our method and prior approaches, demonstrating our design's superiority through comparative experimental results.
    - We explain the **significance of our performance gains** by referencing multiple prior works in visual tracking.
    - We further elaborate on the model sizes commonly used in the single object tracking community and discuss the application of large multi‑modal models in single object tracking.
- **For Reviewer gefL**
    - We highlight that our paper already includes relevant experimental analyses regarding **memory length** and **distractor scenarios**.
    - We further visualize **DSF state changes** under **scale change/fast motion** scenarios, demonstrating DSF's capability to capture dynamic features.
    - We address the reviewer's questions regarding the details of **training** **data composition/ratios**.
    - We clarify that an **ablation study** regarding updating DSF with template integration is already included and analyzed in the appendix of original text.
- **For Reviewer sPSa:**
    - We provide additional **formal mathematical analysis** to clarify our motivation: using the DSF module for **continuous dynamic state updates** rather than directly employing it as a backbone for long-sequence processing. We also elucidate the benefits of combining DSF with MCP.
    - We report the **inference speed** of our method.
    - We provide further analysis of **failure cases** in extremely complex scenarios.
    - We clarify that we have validated the effectiveness of our method **across different baselines**, including DropTrack , which has the same structure as OSTrack.
- **For Reviewer aS26:**
    - We report the **inference speed** of our method and provide corresponding justifications.
    - We further elaborate on the **novelty of our DSF design**, explicitly distinguishing it from prior approaches that utilize SSMs.
    - We emphasize the advantages of our method in long-term tracking scenarios.


We wish everyone a very pleasant day, and we hope our summary will be helpful to you.

---

Best regards,

Authors of Submission 7007

---

### Meta-Review · Area_Chair_4RNs · 2026-01-07

**Summary:**

In the initial review round, the paper received mixed scores (4,6,4,6), with reviewers raising substantial concerns. The primary issues were as follows:

Reviewer 6Uu5: Limited novelty and contribution; only marginal improvements; absence of comparisons with large-model-based tracking approaches.

Reviewer gefL: insufficient ablation studies, lack of qualitative visualization.

Reviewer sPSa: Limited theoretical analysis, heavy reliance on a strong baseline.

Reviewer aS26: lack of actual inference speed, lack of substantial architectural novelty.

Overall, the reviewers' concerns center on limited novelty and empirical/theoretical contributions, insufficient ablations, visualizations, and comparisons (particularly with contemporary large-model-based methods), as well as missing practical details such as inference speed.

**Reviewer Concerns:**

Reviewer 6Uu5: Most concerns—including limited novelty, marginal contributions, and insignificant performance gains—are not adequately addressed. Notably, the rebuttal still lacks comparisons with large-model-based tracking approaches.

Reviewer gefL: All concerns are addressed

Reviewer sPSa: The concern regarding limited theoretical analysis remains unaddressed, as the rebuttal provides no substantial theoretical insights. Other concerns appear resolved.

Reviewer aS26: The concern about lack of substantial architectural novelty is not adequately resolved. The issue of missing inference speed measurements has been addressed.

**Reviewer Scores:**

Reviewer 6Uu5: will keep the initial score (4) or reduce score

Reviewer gefL: will keep the initial score (6)

Reviewer sPSa: will keep the initial score (4) or reduce score

Reviewer aS26: will keep the initial score (6)

Overall, this is a borderline paper. I align with the concerns raised by Reviewers 6Uu5, sPSa, and aS26 regarding the limited novelty, insufficient theoretical analysis, and marginal performance gains of the proposed method. Therefore, I recommend rejection and encourage the authors to substantially strengthen the theoretical foundation, demonstrate clearer architectural novelty and significant empirical improvements.

---

### Decision · Program_Chairs · 2026-01-26

Reject